# Non-cognate immunity proteins provide broader defenses against interbacterial effectors in microbial communities

Abigail Knecht[1,2†‡], Denise Sirias[1†§], Daniel R Utter[3,4#], Karine A Gibbs[1,2*¶]

[1]Departments of Molecular and Cellular Biology, Harvard University, Cambridge, United States; [2]Department of Plant & Microbial Biology, University of California, Berkeley, Berkeley, United States; [3]Departments of Organismic and Evolutionary Biology, Harvard University, Cambridge, United States; [4]Division of Geological and Planetary Sciences, California Institute of Technology, Pasadena, United States

*For correspondence: kagibbs@berkeley.edu

†These authors contributed equally to this work

Present address: ‡Andes Ag, Inc, California, United States; §Tenza, Boston, United States; #Division of Geological and Planetary Sciences, California Institute of Technology, California, United States; ¶Plant & Microbial Biology, University of California, Berkeley, California, United States

## eLife assessment

This study provides **valuable** insights into the specificity and promiscuity of toxic effector and immunity protein pairs. While the work is improved over a previous version, there are still some questions regarding the methodology used to draw certain conclusions, rendering the study somewhat **incomplete**. Nevertheless, this work will likely be of interest to microbiologists and biochemists working with toxin-antitoxin systems and effector-immunity proteins.

## Abstract

Dense microbial communities, like the gut and soil microbiomes, are dynamic societies. Bacteria can navigate these environments by deploying proteins that alter foreign cells' behavior, such as interbacterial effectors. Current models suggest that adjacent sibling cells are protected by an immunity protein, as compared to toxin-antitoxin systems that act only within the effector-producing cell. A prevailing hypothesis is that immunity proteins binding to specific (cognate) protein partners is sufficient to disrupt effector function. Further, there is little-to-no crosstalk with other non-cognate effectors. In this research, we build on sporadic reports challenging these hypotheses. We show that immunity proteins from a newly defined protein family can bind and protect against non-cognate PD-(D/E)XK-containing effectors from diverse phyla. We describe the domains essential for binding and function and show that binding alone is insufficient for protective activity in *Proteus mirabilis*. Moreover, we found that these effector and immunity genes co-occur in individual human microbiomes. These results expand the growing repertoire of bacterial protection mechanisms and the models on how non-cognate interactions impact community structure within complex ecosystems.

## Introduction

Specificity between protein-protein interactions is key for many biological processes, such as metabolism, development, and intercellular signaling. Binding to an incorrect partner or disrupted binding of the specific (cognate) protein can cause a diseased state or cell death (*Kuzmanov and Emili, 2013*). For bacterial social behaviors, cognate protein-protein interactions between cells impact organism fitness and community structure, such as excluding foreign cells (*Cardarelli et al., 2015*). This importance of specificity between cognate partners in social behaviors remains largely unexplored. However, in other contexts, flexible ('promiscuous') binding allows protein partners to retain their interactions when undergoing rapid mutational changes, such as during immune recognition

of viral particles (*Burton et al., 2005*; *Schreiber and Keating, 2011*). Unknown is whether flexible binding between noncognate proteins can occur during bacterial social behaviors and thereby impact microbial communities. An expanded protective function would reveal new bacterial behaviors that influence individual fitness and community structure in dynamic ecosystems.

Microbes often exist within dense, multi-phyla communities, like the human gut microbiome, where they communicate and compete with neighbors. Bacteria can use effector-immunity protein (EI) pairs in these environments to gain advantages (*Russell et al., 2014*; *Speare et al., 2018*). Unlike bacterial toxin-antitoxin (TA) systems in which a single cell produces both toxic and neutralizing proteins, bacteria inject cell-modifying proteins (called interbacterial 'effectors') directly into neighboring cells via several contact-dependent transport mechanisms, including the type VI secretion system (T6SS), type IV secretion system (T4SS), and contact-dependent inhibition (CDI; *Ruhe et al., 2020*; *Sgro et al., 2019*). Clonal siblings produce the matching immunity protein that modifies the effector's activity. For lethal effectors, both clonal and non-clonal cells are negatively impacted when binding is disrupted or absent (*Russell et al., 2014*). These interactions between EI pairs can shape community composition by changing bacterial fitness.

The interaction specificity between matching EI pairs has historically defined immunity protein protection, but recent studies raise doubts. Of note, EI pairs interact within a neighboring cell which creates unique restrictions for both their protection mechanisms and their evolution (*Jurénas et al., 2022*). Currently, the predominant model is that T6SS-associated EI pairs act like a tumbler lock-and-key, where each effector protein has a single cognate partner (*Hersch et al., 2020*). Immunity proteins bind their cognate effectors, often at the active site, to neutralize effector activity (*Benz and Meinhart, 2014*; *Hagan et al., 2023*). However, experiments with engineered proteins reveal that small amino acid sequence changes to an immunity protein can allow it to bind effectors other than its cognate partner (*Alteri et al., 2017*; *Levin et al., 2009*). Also, the T6SS-associated effector and immunity proteins from *Salmonella enterica* subsp. *enterica* serovar Typhimurium and *Enterobacter cloacae*, which are phylogenetically close, bind each other in vitro and protect against the other in vivo (*Zhang et al., 2013*). Another example is Tde1 and Tdi1. Homologous Tdi1 immunity proteins lacking a cognate effector (i.e. 'orphans') bound and protected against the effector from a different organism (*Bosch et al., 2023*). These studies indicate that the widely used tumbler lock-and-key model does not account for the potential breadth of immunity protein protection.

We studied an EI pair in *Proteus mirabilis* to examine this prevailing model. This opportunistic pathogen resides in human and animal guts and can cause recurrent and persistent urinary tract infections (*Schaffer and Pearson, 2015*). *P. mirabilis* encodes two T6SS-dependent EI pairs (one lethal and one non-lethal) that impact collective motility and relative fitness (*Saak et al., 2017*; *Wenren et al., 2013*). For the lethal EI pair, previously termed Idr (*Wenren et al., 2013*), the molecular functions remained unknown. Here, we characterized this EI pair and determined the critical residues for activity, leading to the identification of two protein families. We showed that proteins in the immunity protein family bind non-cognate effectors produced by bacteria from different phyla and result in altered population structures. Structure-function assays revealed that a conserved region within the C-terminus of the immunity proteins is necessary to neutralize the *P. mirabilis* effector protein. Further, we found that the flexible EI pairs from various phyla naturally co-occur in individual human microbiomes. These findings provide compelling evidence for cross-protection and support a critical revision of the model for EI pairs, particularly in consideration of ecological significance.

## Results

### RdnE is a DNA nuclease and seeds a PD-(D/E)XK subfamily

To compete against other strains, *P. mirabilis* strain BB2000 requires both the *idrD* gene and the T6SS, suggesting that the *idrD*-encoded protein functions as a T6SS-associated effector (*Saak et al., 2017*). The *idrD* gene contains an Rhs region within its N-terminus. Many Rhs-containing effectors often contain an enzymatic domain in the C-terminus (*Koskiniemi et al., 2013*; *Ma et al., 2017*). As a result, we investigated the function of the final 138 amino acids at IdrD's C-terminus, now renamed 'RdnE' for recognition DNA nuclease effector. We measured bacterial growth using a strain derived from BB2000 that has disruptions in its native *idrD* and downstream genes (*Wenren et al., 2013*). This *P. mirabilis* culture had 1000 fewer cells per mL when engineered to overproduce RdnE in trans than

the negative control containing the parent empty vector (*Figure 1A*). An equivalent growth pattern occurred in *Escherichia coli* cells under the same conditions (*Figure 1—figure supplement 1*). Thus, RdnE was lethal in vivo.

RdnE's initial 86 amino acids contain a PD-(D/E)XK motif, which is suggestive of nucleotide degradation. The PD-(D/E)XK superfamily includes proteins with broad functions, including effectors that degrade DNA or RNA (*Jana et al., 2019*; *Kosinski et al., 2005*; *Yadav et al., 2021*). Three residues in the catalytic site—D, D/E, and K—are required for activity (*Steczkiewicz et al., 2012*). Therefore, we changed the corresponding residues in RdnE (D39, E53, and K55) to alanine, separately and together. *P. mirabilis* producing these mutant proteins showed growth equivalent to the negative control lacking RdnE (*Figure 1A*). We also saw that *E. coli* cells that were producing RdnE had morphologies that were indicative of DNA damage or stress, consistent with an SOS response (*Friedberg et al., 2005*; *Kreuzer, 2013*). These cells were elongated, and the DAPI-stained DNA was distributed irregularly within the cells (*Figure 1—figure supplement 1*). Cells producing a D39A mutant (RdnE$_{D39A}$) largely did not have this appearance, although a few elongated cells remained, suggesting that the D39A mutant retained partial activity (*Figure 1—figure supplement 1*). Therefore, the PD-(D/E)XK motif was essential for cell death.

The importance of the PD-(D/E)XK motif for activity suggested that RdnE was a nuclease, but defining its molecular target required direct analysis. Due to its lethality in *P. mirabilis* and *E. coli* cells, we synthesized RdnE with a C-terminus FLAG epitope tag using in vitro translation, which resulted in nanogram quantities (*Figure 1—figure supplement 1*). We added phage lambda DNA (methylated or unmethylated) to progressively higher RdnE protein concentrations and then performed agarose gel electrophoresis analysis. Degradation of lambda DNA occurred in the presence of RdnE, regardless of the DNA methylation state (*Figure 1B*). The RdnE$_{D39A}$ construct caused a slight reduction in lambda DNA, while the negative control showed no DNA loss (*Figure 1B*). RdnE also caused a reduction in plasmid DNA, indicating it has endonuclease activity (*Figure 1—figure supplement 1*). These results revealed that RdnE caused DNA degradation in vitro in a PD-(D/E)XK-dependent manner.

RdnE appeared to have two different domains, as a region directly follows the PD-(D/E)XK motif. A two-domain architecture is similar to that described for DNases (*Lowey et al., 2020*; *Schiltz et al., 2019*). Yet, the PD-(D/E)XK domain could also be sufficient for DNase activity of some effectors such as PoNe-containing DNases (*Hespanhol et al., 2022*; *Jana et al., 2019*). Therefore, we examined whether RdnE's PD-(D/E)XK motif was sufficient for DNA degradation or whether both domains were required for activity. We made independent deletions of each potential RdnE domain. One construct deleted the first alpha helix without disturbing the catalytic residues; the other deleted the region after the PD-(D/E)XK motif, which we termed 'region 2'. The resulting proteins, produced via in vitro translation, were assayed for DNase activity as described above. The truncated proteins resulted in no loss of lambda DNA (*Figure 1C*), indicating that both domains were necessary for degradation activity.

We next asked whether RdnE homologs also act as DNA nucleases. A bioinformatics search revealed the closest RdnE homolog outside of *Proteus* was found in the Actinobacteria, *Rothia dentocariosa* C6B. *Rothia* species are inhabitants of the normal oral flora, dwelling in biofilms within the human oral cavity and pharynx (*Wilbert et al., 2020*). The two RdnE proteins (^Proteus^RdnE and ^Rothia^RdnE) share approximately 55% amino acid sequence identity, mostly within the PD-(D/E)XK domain; they have similar predicted secondary structures (*Figure 1D*). Given this, we hypothesized that ^Rothia^RdnE also acted as a DNA nuclease.

We analyzed ^Rothia^RdnE for PD-(D/E)XK-dependent DNA nuclease activity by producing it and a predicted null mutant, ^Rothia^RdnE$_{D39A}$, using in vitro translation. Samples containing the ^Rothia^RdnE$_{D39A}$ protein or a negative control had similar DNA levels (*Figure 1E*). By contrast, samples with the wild-type ^Rothia^RdnE protein showed a loss of lambda DNA regardless of methylation state, indicating that ^Rothia^RdnE also had DNA nuclease activity (*Figure 1E*). Given that region 2 was necessary for activity in ^Proteus^RdnE but has greater amino acid sequence diversity than the PD-(D/E)XK domain between the two proteins, we queried whether domains from foreign organisms could complement one another. We exchanged region 2 between the ^Proteus^RdnE and ^Rothia^RdnE sequences and assayed for nuclease activity. The hybrid proteins degraded lambda DNA, unlike the negative control (*Figure 1F*), demonstrating the cross-phyla protein domains could complement one another. Altogether, these findings demonstrated that these RdnE proteins form a PD-(D/E)XK-containing DNA nuclease subfamily. This

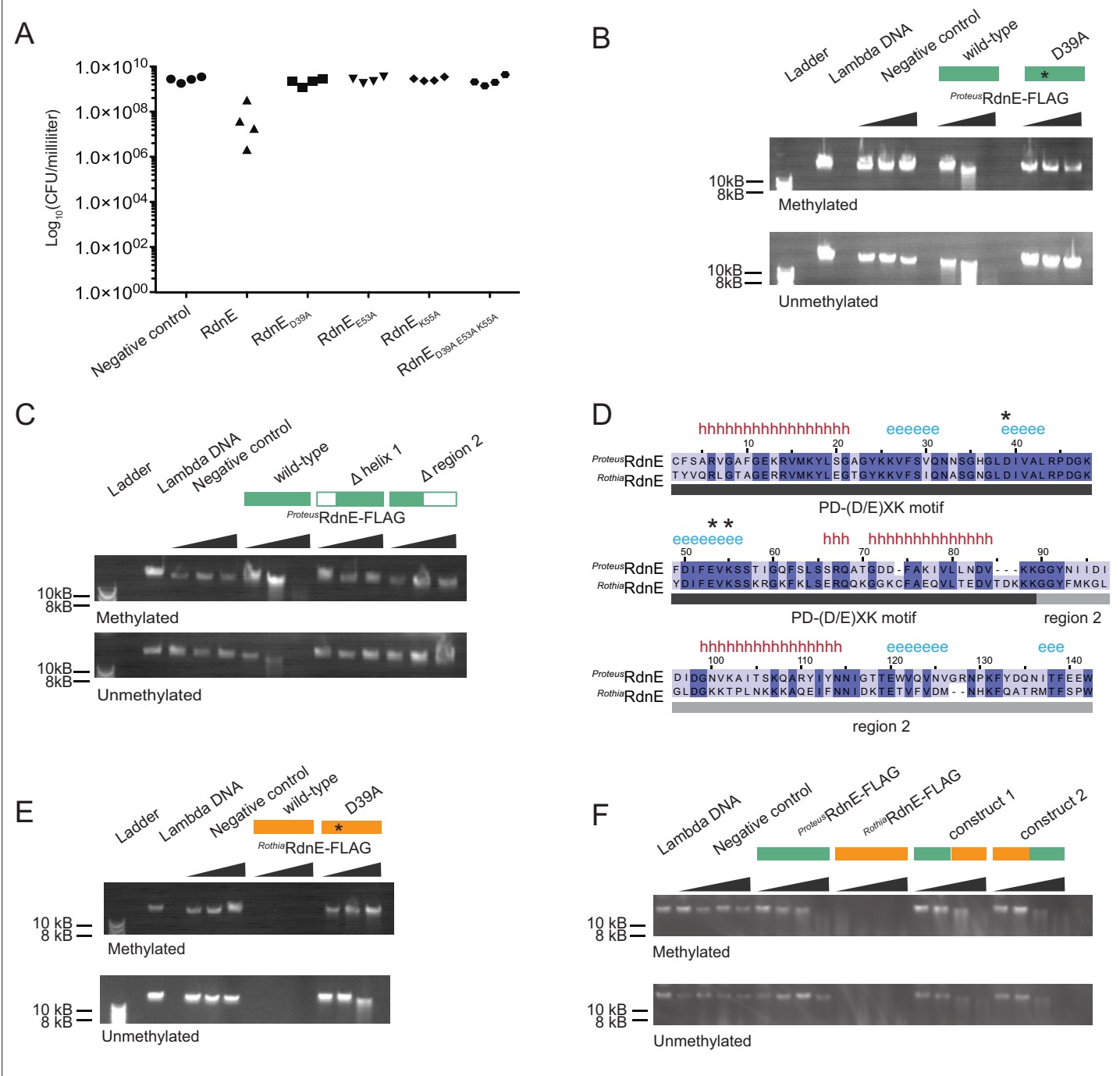

**Figure 1.** RdnE homologs act as DNA endonucleases and contain interchangeable domains. (**A**) Cell viability (colony forming units [CFU] per mL) after protein production in swarms of *P. mirabilis* strain *idrD\**, which does not produce RdnE and RdnI. Cells produced GFPmut2, RdnE, or mutant variants in the predicted PD-(D/E)XK motif: D39A, E53A, K55A, or all. (**B**) In vitro DNA degradation assay for ^Proteus^RdnE. Increasing concentrations of a negative control, ^Proteus^RdnE-FLAG, or ^Proteus^RdnE~D39A~-FLAG were incubated with methylated or unmethylated lambda DNA (48,502 bp) and analyzed by gel electrophoresis. Plasmid DNA degradation is in *Figure 1—figure supplement 1*. (**C**) In vitro DNA degradation assay for domain deletions of ^Proteus^RdnE. The first construct removed the first alpha helix without disturbing the catalytic residues, and the second construct contained the PD-(D/E)XK motif and removed region 2. Increasing concentrations were analyzed as in (**B**). (**D**) Multiple sequence alignment between *P. mirabilis* and *R. dentocariosa* RdnE sequences. The black bar marks the PD-(D/E)XK motif, and the gray bar marks the variable region 2 domain. Conserved residues are highlighted in dark blue. Secondary structure predictions identified using Ali2D (h for alpha helix, e for beta sheet); the catalytic residues (stars) are noted above the alignment. (**E,F**) In vitro DNA degradation assay and analysis as in (**B**). (**E**) Increasing concentrations of either a negative control, ^Rothia^RdnE-FLAG,

*Figure 1 continued on next page*

Figure 1 continued

or $^{Rothia}$RdnE$_{D39A}$-FLAG. (**F**) The PD-(D/E)XK motifs were swapped between the $^{Rothia}$RdnE (orange bar) and the $^{Proteus}$RdnE (green bar) sequences and compared to the wild-type RdnE proteins.

The online version of this article includes the following source data and figure supplement(s) for figure 1:

**Source data 1.** The full gels of the data in *Figure 1B, C, E and F*.

**Source data 2.** The individual, original gel scans for the data in *Figure 1B, C, E and F*.

**Figure supplement 1.** RdnE, an endonuclease, is lethal in *Escherichia coli* and cuts plasmid DNA.

**Figure supplement 1—source data 1.** The full gels of the data in *Figure 1—figure supplement 1C and D*.

**Figure supplement 1—source data 2.** It contains individual, original gel scans for the data in *Figure 1—figure supplement 1C and D*.

conclusion is also consistent with recent literature showing that RdnE-containing proteins (formerly IdrD-CT [*Sirias et al., 2020*]) form their own sub-clade within other PD-(D/E)XK-containing nucleases (*Hespanhol et al., 2022*).

## RdnI binds and neutralizes RdnE

As effectors have cognate immunity proteins that are often located adjacently on the chromosome, we hypothesized that *rdnI* (formerly "*idrE*"), which is the gene directly downstream of *rdnE* in *P. mirabilis* (*Figure 2A*), encodes the cognate immunity protein. RdnI did not have defined domains, and its function was unknown. We assessed RdnI's activity using microscopic and cell growth analysis. Swarming *P. mirabilis* cells are normally elongated with DAPI-stained DNA found along the cell body (*Figure 2B*). By contrast, swarming cells producing RdnE in trans did not elongate, had a reduced DAPI signal, and had an accumulation of misshapen cells (*Figure 2B*). Cell shape and DNA-associated fluorescence levels returned to normal when cells concurrently produced the RdnE and RdnI proteins (*Figure 2B*). RdnI production also rescued cell growth in *E. coli* cells producing RdnE (*Figure 2—figure supplement 1*). These data suggested that RdnI inhibits RdnE's lethality.

We next evaluated whether RdnI provided protection against injected RdnE within mixed communities similar to native ecosystems. We used well-established swarm competition assays, which combine one-to-one mixtures of *P. mirabilis* strains to measure dominance in two-dimensional population structures (*Wenren et al., 2013*). The control strain was wild-type strain BB2000 (herein called 'BB2000'), which naturally produces RdnE and RdnI. The other was strain ATCC29906, which does not naturally produce RdnE and RdnI. These two strains formed a visible boundary between swarming monoculture colonies (*Figure 2C*). The mixed-strain colony merged with BB2000 in one-to-one competitions, demonstrating BB2000's dominance in the two-dimensional population structure (*Figure 2C*). A similar outcome was seen when ATCC29906 produced a vector-encoded Green Fluorescent Protein (GFPmut2) under the *fla* promoter, which results in constitutive gene expression in swarming *P. mirabilis* cells (*Belas et al., 1991*; *Jansen et al., 2003*). However, BB2000 did not outcompete ATCC29906 engineered to produce vector-encoded RdnI with a C-terminal Strep-tag II epitope tag ('RdnI-StrepII') under the *fla* promoter; this is visible in the merging of the mixed-strain colony with ATCC29906 (*Figure 2C*). Thus, RdnI protected cells against injected RdnE within mixed communities.

Based on the prevailing EI model, we predicted that a cognate EI pair should bind to one another, which we evaluated in vivo and in vitro. We used the attenuated mutant (RdnE$_{D39A}$-FLAG) for these assays because producing the wild-type RdnE protein kills cells. For in vivo analysis, we used bacterial two-hybrid assays (BACTH) in which the reconstitution of the T18 and T25 fragments of adenylate cyclase results in the colorimetric change to blue in the presence of the substrate, X-gal (*Battesti and Bouveret, 2012*; *Karimova et al., 1998*). Constructed vectors contained genes for RdnE$_{D39A}$-FLAG, RdnI-StrepII, or GFPmut2 on the C-termini of the T18 or the T25 fragment. When the reporter strain produced RdnE$_{D39A}$-FLAG or RdnI-StrepII with GFPmut2, the resultant yellow color was equivalent to when X-gal was absent (*Figure 2D*). There was also minimal color change when an individual protein was produced on both fragments (*Figure 2D*). However, the reporter strains made blue colonies when X-gal was present, and the cells concurrently produced RdnE$_{D39A}$-FLAG and RdnI-StrepII. These results indicated that RdnE and RdnI bind to each other in vivo.

We used batch in vitro co-immunoprecipitation assays to confirm the in vivo binding result. Separate *E. coli strains* produced either RdnE$_{D39A}$-FLAG or had a negative control, exogenous FLAG-BAP (*E. coli* bacterial alkaline phosphatase with a FLAG epitope tag) added to cell lysate. An anti-FLAG

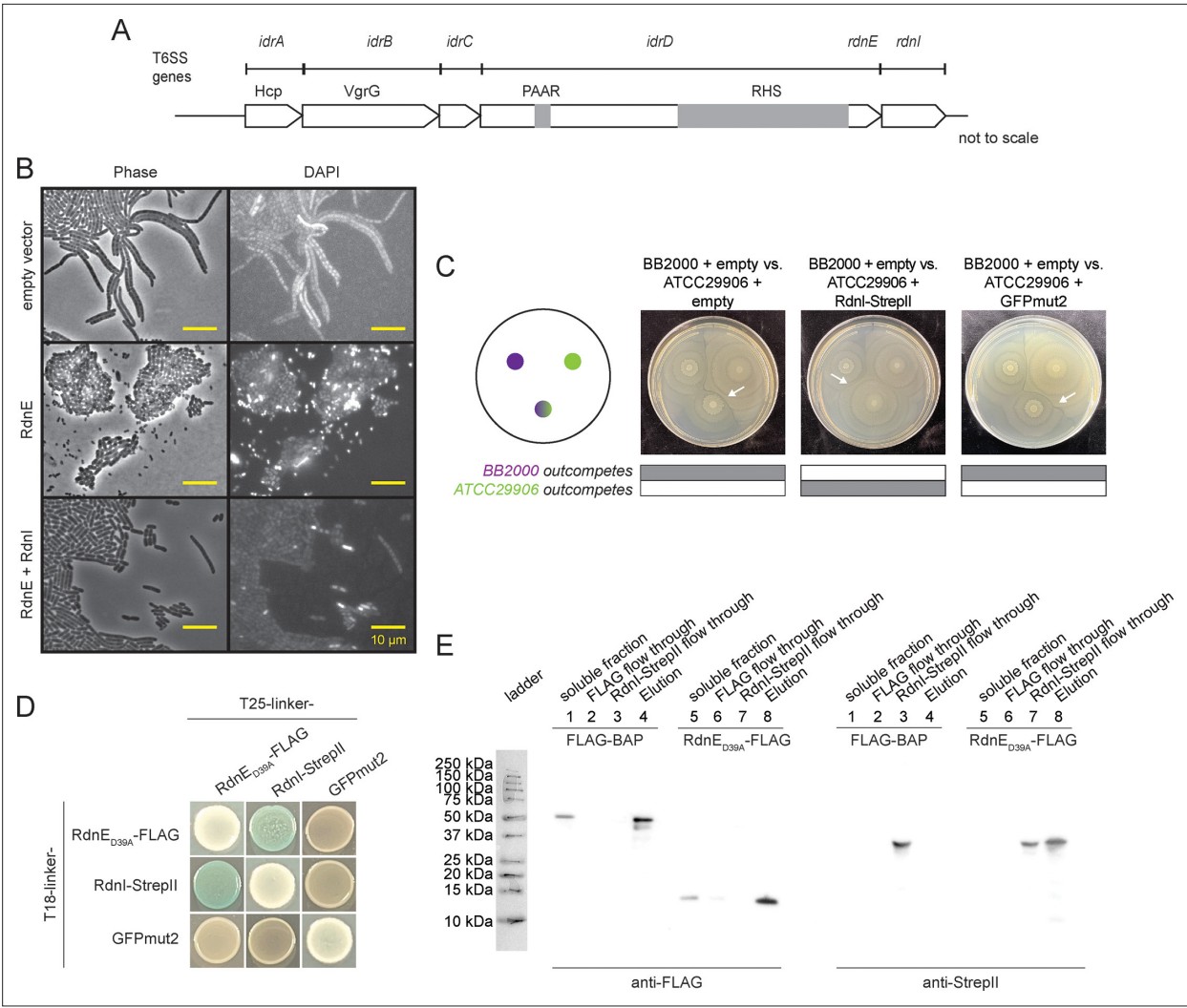

**Figure 2.** RdnI binds to and protects against RdnE in vivo and in vitro. (**A**) Domain architecture for the *idr* locus in *P. mirabilis* strain BB2000. At the top are genes with Pfam domains listed below them. Gray boxes denote PAAR and Rhs domains in the N-terminal region of the full-length IdrD protein. (**B**) Micrographs of *P. mirabilis* strain *idrD\** cells carrying an empty vector, a vector for producing RdnE, or a vector for producing RdnE and RdnI. DNA was visualized by DAPI stain. Phase, left; fluorescence, right. (**C**) Swarm competition assay of wild-type *P. mirabilis* strain BB2000 (donor) competed against the vulnerable target, which is *P. mirabilis* strain ATCC29906 carrying an empty vector, a vector for producing RdnI-StrepII, or a vector for producing GFP, both under the *fla* promoter. Left: schematic of swarm competition assay where top left colony is BB2000, top right colony is ATCC29906 with its vector cargo, and bottom colony is a 1:1 mixture of BB2000 and ATCC29906 with its vector cargo. Gray boxes underneath indicate whether BB2000 (top) or ATCC29906 (bottom) dominate in the 1:1 mixture and white arrows point to a boundary line that forms between different strains. (**D**) Bacterial two-hybrid (BACTH) assay with RdnE$_{D39A}$-FLAG, RdnI-StrepII, and GFPmut2. The colorimetric change was discerned in the presence of the substrate X-gal and inducer IPTG. (**E**) An anti-FLAG batch co-immunoprecipitation of RdnE$_{D39A}$-FLAG and RdnI-StrepII. RdnE$_{D39A}$-FLAG or exogenous FLAG-BAP (soluble fraction) was incubated with anti-FLAG resin (FLAG flow through). RdnI-StrepII was then added to the resin (RdnI-StrepII flow through). Any proteins bound to resin were eluted with FLAG-peptide (Elution) and analyzed by anti-FLAG and anti-StrepII western blots.

The online version of this article includes the following source data and figure supplement(s) for figure 2:

**Source data 1.** It contains the full gels of the data in *Figure 2E*.

**Source data 2.** It contains the individual, original gel scans for the data in *Figure 2E*.

**Figure supplement 1.** RdnI offers protection against and binds to RdnE.

**Figure supplement 1—source data 1.** It contains the full gels of the data in *Figure 2—figure supplement 1B*.

**Figure supplement 1—source data 2.** It contains the individual, original gel scans for the data in *Figure 2—figure supplement 1B*.

western blot showed both FLAG-BAP (~50 kDa) and RdnE$_{D39A}$-FLAG (~17 kDa) in the soluble and elution fractions. RdnI-StrepII eluted with RdnE$_{D39A}$-FLAG but not the negative control (*Figure 2E*). The western blot results corresponded with the Coomassie blue-stained gels (*Figure 2—figure supplement 1*). Overall, our data showed that *Proteus* RdnE and RdnI form a cognate EI pair with impacts on population structure. Questions about their prevalence among bacteria and their ecological relevance remained.

## Expansion of the RdnE and RdnI protein families revealed similar gene architecture and secondary structures

Gene neighborhood analysis can guide homology inference and protein comparisons. We conducted consecutive searches with BLAST (*Altschul et al., 1990*) and HMMER (*Eddy, 2009*) to identify sequences that encoded proteins with high similarity to RdnE and RdnI (*Figure 3—figure supplement 1*). The final list contained 21 EI pairs from a variety of phyla that are located adjacently in their respective genomes (*Table 1*). Although the genes surrounding these putative EI pairs differed, many shared mobile-associated elements, such as Rhs sequences or other similar peptide-repeat sequences (*Figure 3A*). Several gene neighborhoods had secretion-associated genes, such as the T6SS-associated *vgrG/tssI* gene and the CDI-associated *cdiB* gene. A few also included putative immunity proteins from other reported families, like immunity protein 44 (Pfam15571) in *Taylorella asinigenitalis* MCE3 and immunity protein 51 (Pfam15595) in *Chryseobacterium populi* CF314. Notably, these organisms varied widely in origin and residence. Some were from the soil rhizosphere (*Pseudomonas ogarae* and *C. populi*) and others from the human microbiome (*P. mirabilis*, *R. dentocariosa*, and *Prevotella jejuni*; *Figure 3A*). The prevalence of these genes across the phylogenetic tree (*Figure 3B*) and the presence of secretion-associated loci in the gene neighborhoods suggested a role in cell-cell interactions and potentially community structure.

Given the diversity in species, we next examined the relationship between the various RdnE- and RdnI-like proteins and whether there was syntony between these proteins given that they are encoded adjacently on each identified genome. We constructed maximum likelihood trees to examine the relationship between the identified RdnE- and RdnI-like proteins. The RdnE and RdnI trees diverged from the species tree (*Figure 3B*). However, overall, the arrangement of RdnE- and RdnI-like proteins within the maximum likelihood trees was similar and showed syntony (*Figure 3—figure supplement 1*), although small differences were present (*Figure 3C*). For example, *P. mirabilis* and *R. dentocariosa* proteins shared more similarities than to those from more closely related genera. These results are consistent with the potential horizontal gene transfer reported for other EI pairs (*Ruhe et al., 2020*).

Given these results, we reasoned that the domain architectures and amino acid diversity could reveal functions for the two families. When we examined the predicted secondary structures of the RdnE-like proteins, they were conserved despite differences in the amino acid sequences (*Figure 3D*, *Figure 3—figure supplement 1*). The RdnE proteins showed two distinct domains, as with the *Proteus* and *Rothia* results (*Figure 1*): the PD-(D/E)XK region followed by a sequence variable region (region 2). Further, the AlphaFold2-generated (*Jumper et al., 2021*; *Mirdita et al., 2022*) predicted tertiary structures were consistent with PD-(D/E)XK folds (three β-sheets flanked by two α-helices, α/β/α) found in other proteins (*Figure 3E*, *Figure 3—figure supplement 2*; *Steczkiewicz et al., 2012*). These findings suggested that domains in RdnE are conserved across diverse phyla and further confirm that the sequences seed a distinct PD-(D/E)XK subfamily.

While immunity proteins within a family have diverse overall amino acid sequences, conserved secondary structures and some conserved residues are common in some immunity protein families. Indeed, they are often used to characterize these families (*Zhang et al., 2012*). We found that while the RdnI proteins shared minimal primary amino acid sequence identity, they were predicted to contain several α-helices (*Figure 3D*, *Figure 3—figure supplement 1*) and had similar AlphaFold2-predicted tertiary structures (*Figure 3E*, *Figure 3—figure supplement 2*). We also discovered a region with three alpha-helices and several conserved residues, which we named the 'conserved motif' (*Figure 3D*). The RdnI conserved motif might be a key domain for seeding this novel immunity protein family.

## Binding flexibility in RdnI allows for cross-species protection

We deployed a structure-function approach to determine the conserved motif's role in $^{Proteus}$RdnI's activity. Analysis using AlphaFold2 (*Jumper et al., 2021*; *Mirdita et al., 2022*) and Consurf (*Ashkenazy*

**Table 1.** RdnE and RdnI homolog species.

| Genus species strain | Phylum | Isolation Location | RdnE JGI unique ID | RdnI JGI unique ID |
|---|---|---|---|---|
| *Acinetobacter baumannii* BJAB0715 | Proteobacteria | Fresh water | 2562302616 | 2562302617 |
| *Acinetobacter baumannii* XH858 | Proteobacteria | Human sputum | 2688809281 | 2688809282 |
| *Burkholderia* sp. TSV86 | Proteobacteria | Water | 2766166119 | 2766166118 |
| *Cellulophaga baltica* 18 | Bacteroidota | Water | 2815949879 | 2815949878 |
| *Chryseobacterium indologenes* NBRC 14944 | Bacteroidota | Human trachea | 2565567985 | 2565567984 |
| *Chryseobacterium populi* CF314 | Bacteroidota | Soil rhizosphere | 2511231970 | 2511231971 |
| *Chryseobacterium* sp. IHB B 17019 | Bacteroidota | Soil undefined subtype | 2688963654 | 2688963655 |
| *Cronobacter turicensis* 564 | Proteobacteria | Human undefined subtype | 2532469359 | 2532469360 |
| *Cronobacter turicensis* z3032 | Proteobacteria | Human blood culture | 646327905 | 646327906 |
| *Endozoicomonas numazuensis* DSM 25634 | Proteobacteria | Marine sponge | 2574519540 | 2574519541 |
| *Paenibacillus elgii* M63 | Firmicutes | Hot spring | 2744846532 | 2744846531 |
| *Paenibacillus* sp. Aloe-11 | Firmicutes | Soil rhizosphere | 2549870597 | 2549870598 |
| *Prevotella jejuni* CD3:33 | Bacteroidota | Human intestine biopsy | 2804797915 | 2804797916 |
| *Prevotella* sp. C561 | Bacteroidota | Human respiratory tract | 2514485316 | 2514485315 |
| *Prevotella* sp. F0108 | Bacteroidota | Human oral cavity | 647936965 | 647936966 |
| *Proteus mirabilis* BB2000 | Proteobacteria | Human intestine biopsy | 2546214711 | 2546214712 |
| *Pseudomonas ogarae* F113 | Proteobacteria | Soil rhizosphere | 2511826458 | 2511826457 |
| *Pseudomonas syringae pv. Coriandricola* ICMP 12471 | Proteobacteria | Undefined | 2714543877 | 2714543878 |
| *Rothia dentocariosa* C6B | Actinobacteriota | Human oral cavity | 2611822673 | 2611822674 |
| *Tannerella forsythia* ATCC 43037 | Bacteroidota | Human oral cavity | 2512371376 | 2512371377 |
| *Taylorella asinigenitalis* MCE3 | Proteobacteria | Mammal reproductive system | 2511725471 | 2511725470 |

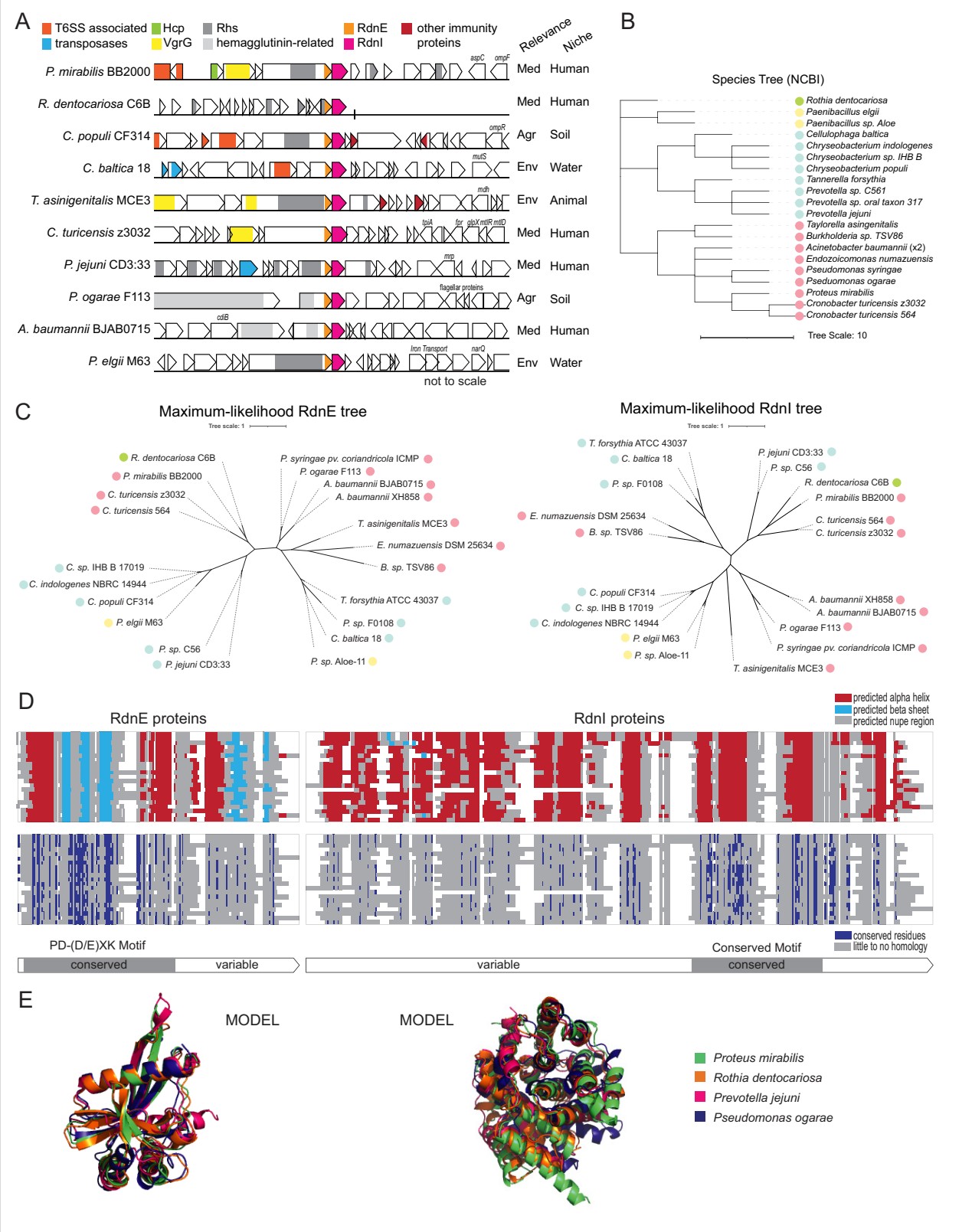

**Figure 3.** RdnE and RdnI protein families share conserved residues and predicted structures. (**A**) Gene neighborhoods for RdnE and RdnI homologs. Listed are gene neighborhoods, relevance, and niche, which we identified using IMG/M from the Joint Genomics Institute. Colors highlight conserved function/genes (not to scale). (Agr: Agriculture, Med: Medical, Env: Environmental), and the site of isolation. (**B**) Phylogenetic tree based on NCBI taxonomy. Scale is located below the graph. The colored circles represent phyla (green: Actinobacteriota; yellow: Firmicutes; blue: Bacteroidota; pink:

*Figure 3 continued on next page*

*Figure 3 continued*

Proteobacteria). (**C**) Unrooted maximum likelihood trees of the RdnE (left) and RdnI (right) homologs. Trees were created with RaxML (*Kozlov et al., 2019*), and the scale is annotated below. The colored circles represent phyla (same as in B). (**D**) Protein alignments overlaid with either predicted secondary structures (top) or conserved residues (bottom) of the RdnE and RdnI homologs. MUSCLE alignments (*Edgar, 2004*) are highlighted by secondary structures (red: alpha helices, light blue: beta sheets), or conserved residues (dark blue). White represents gaps in the protein alignment. The bars below mark the predicted conserved and variable domains. (**E**) Alignments of AlphaFold2 predictions for RdnE and RdnI sequences from *P. mirabilis* (green), *R. dentocariosa* (orange), *P. jejuni* (magenta), and *P. ogarae* (dark blue). Structures were generated using ColabFold (*Mirdita et al., 2022*) and aligned using PyMol.

The online version of this article includes the following figure supplement(s) for figure 3:

**Figure supplement 1.** RdnE and RdnI protein families show conserved structures.

**Figure supplement 2.** AlphaFold2 predictions for RdnE and RdnI homologs.

*et al., 2016*) revealed seven highly conserved residues within this region; four of these (Y197, H221, P244, E246) clustered together within the AlphaFold2 structure and are identical between sequences (*Figure 4A*). In a sequence-optimized (SO) RdnI, we independently changed each of these four residues to alanine and discovered that each alanine-substituted variant behaved like the wildtype and inhibited RdnE activity (*Figure 4—figure supplement 1*). We then replaced all seven residues (Y197, S235, K258, and the original four) with alanine ($^{Proteus}$RdnI$_{7mut}$-StrepII) and found that, unlike wild-type $^{Proteus}$RdnI-StrepII, this construct was not protective in swarm competition assays (*Figure 4B*). However, the $^{Proteus}$RdnI$_{7mut}$-StrepII mutant still bound RdnE$_{D39A}$-FLAG in bacterial two-hybrid assays (*Figure 4C*). Therefore, the seven residues in the conserved motif are critical for RdnI's neutralizing function but dispensable for binding RdnE.

Given that the conserved motif and nearby regions are likely involved in protective activity, we queried for potential functions in the remainder of the RdnI protein. We engineered variants that were either (1) the first 85 amino acids, (2) amino acids 150–305, which contained an intact conserved motif, or (3) amino acids 235–305, which contained the last alpha helix of the conserved motif (*Figure 3—figure supplement 1*). None of these constructs protected against RdnE's lethality in vivo during the swarm competition assay (*Figure 4D*), demonstrating that the entire protein is likely essential for function. However, the variant containing the first 85 amino acids of $^{Proteus}$RdnI was the only construct to bind $^{Proteus}$RdnE, indicating that the N-terminal region is sufficient for binding between this *P. mirabilis* EI pair (*Figure 4E*). Thus, our data suggests that binding is necessary but not sufficient for neutralization. Also, the inhibitory activity might reside within the second half of RdnI. As the prevailing model defines cognate-specificity by binding activity, our structure-function results for RdnE (*Figure 1F*) and RdnI (*Figure 4E*) suggest that this model does not fully explain the complex interactions between effectors and immunity proteins.

Therefore, we explored the relationship between non-cognate RdnE and RdnI proteins from various phyla. We first asked whether non-cognate RdnI immunity proteins could protect against injected $^{Proteus}$-RdnE. Using the swarm competition assays, we competed BB2000 against ATCC29906 engineered to produce vector-encoded RdnI homologs from *P. mirabilis*, *R. dentocariosa*, *P. jejuni*, or *P. ogarae* ($^{Proteus}$RdnI-StrepII, $^{Rothia}$RdnI-StrepII, $^{Prevotella}$RdnI-StrepII, and $^{Pseudomonas}$RdnI-StrepII, respectively) under the *fla* promoter (*Figure 2*). BB2000 dominated the swarm when ATCC29906 produced GFPmut2, $^{Prevotella}$RdnI-StrepII, or $^{Pseudomonas}$RdnI-StrepII (*Figure 4F*). However, ATCC29906 outcompeted BB2000 when making $^{Proteus}$RdnI-StrepII or $^{Rothia}$RdnI-StrepII (*Figure 4F*). Expression levels of the transgenic RdnI proteins in ATCC29906 were similar (*Figure 4—figure supplement 2*). Further, the RdnI immunity proteins from *Proteus* and *Rothia* consistently bound $^{Proteus}$RdnE$_{D39A}$ in the in vivo and in vitro assays; the *Prevotella* and *Pseudomonas* variants did not (*Figure 4G*, *Figure 4—figure supplement 3*). The binding to and protection of $^{Rothia}$RdnI against $^{Proteus}$RdnE demonstrated that cross-protection between non-cognate EI pairs from different phyla is possible, provides a fitness benefit during competition, and influences community structure.

While overall the EI pairs showed syntony with each other (*Figure 3—figure supplement 1*), amino acid changes can have critical impacts on whether there are specific or flexible interactions between non-cognate protein pairs (*Schreiber and Keating, 2011*). Therefore, we next evaluated which region(s) of RdnI contributes to cross-protection. We first moved the conserved motif of the three foreign RdnI homologs into $^{Proteus}$RdnI-StrepII and measured neutralizing activity using swarm competition assays and binding activity using BACTH. The conserved motifs from *Rothia* and *Prevotella* were

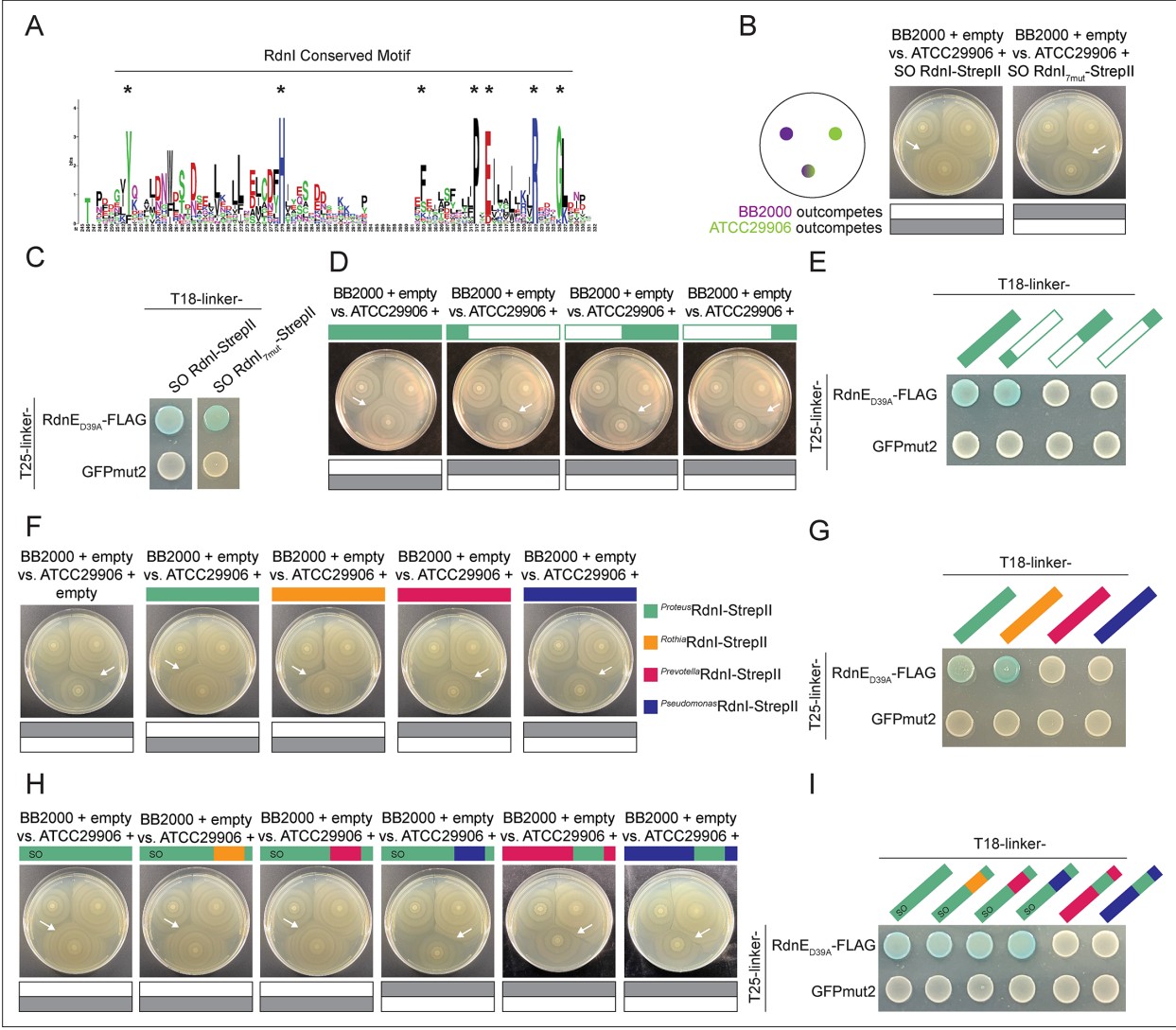

**Figure 4.** The RdnI protein family can offer cross-protection due to an interchangeable conserved domain that is critical for function. (**A**) Sequence logo of the RdnI's conserved motif. Stars indicate the seven analyzed residues. (**B**) Swarm competition assay with ATCC29906 producing either RdnI-StrepII or RdnI$_{7mut}$-StrepII, which contains mutations in all seven conserved residues. We used a sequence-optimized (SO) RdnI protein that had a higher GC% content and an identical amino acid sequence for ease of cloning. Left: schematic of swarm competition assay as in *Figure 2*. Gray boxes indicate which strain dominated over the other. White arrows point to the boundary formed between different strains. (**C**) BACTH assay of RdnE$_{D39A}$-FLAG with SO RdnI-StrepII or RdnI$_{7mut}$-StrepII. GFPmut2 was used as a negative control. (**D**) Swarm competition assay with ATCC29906 expressing either the wild-type RdnI or a RdnI truncation. The three truncations were in the first alpha helix (amino acids 1–85), the second half of RdnI (amino acids 150–305), and the end of the protein (amino acids 235–305). (**E**) BACTH assay of RdnE$_{D39A}$-FLAG with wild-type RdnI and the three RdnI truncations. (**F**) Swarm competition assay with ATCC29906 expressing foreign RdnI proteins. (**G**) BACTH assay of RdnE$_{D39A}$-FLAG with each of the foreign RdnI proteins. GFPmut2 was used as a negative control. (**H**) Swarm competition assay with ATCC29906 producing SO RdnI with swapped conserved motifs. (**I**) BACTH assay of RdnE$_{D39A}$-FLAG with SO RdnI with swapped conserved motifs. Colored bars denote RdnI-StrepII proteins from *P. mirabilis* (green), *R. dentocariosa* (orange), *P. jejuni* (magenta), or *P. ogarae* (dark blue).

The online version of this article includes the following source data and figure supplement(s) for figure 4:

**Figure supplement 1.** Single mutations in the RdnI conserved motif do not alter protective function.

**Figure supplement 2.** RdnI protein levels are similar under constitutive *fla* promoter in *P. mirabilis*.

**Figure supplement 2—source data 1.** It contains the full gels of the data in *Figure 4—figure supplement 2*.

**Figure supplement 2—source data 2.** It contains the individual, original gel scans for the data in *Figure 4—figure supplement 2*.

**Figure supplement 3.** Anti-FLAG co-IPs reveal mixed binding results for foreign immunity protein Anti-FLAG co-immunoprecipitation assay between $^{Proteus}$RdnE$_{D39A}$-FLAG and the RdnI-StrepII proteins from *P. mirabilis, R. dentocariosa, P. jejuni,* or *P. ogarae*.

**Figure supplement 3—source data 1.** The full gels for the data in *Figure 4—figure supplement 3*.

**Figure supplement 3—source data 2.** The individual, original gel scans for the data in *Figure 4—figure supplement 3*.

sufficient to preserve $^{Proteus}$RdnI's neutralizing (*Figure 4H*) and binding functions (*Figure 4I*). However, the conserved motif from *Pseudomonas* was not sufficient to neutralize $^{Proteus}$RdnE (*Figure 4H*), even though the construct still bound $^{Proteus}$RdnE$_{D39A}$ (*Figure 4I*). We then moved the *Proteus* conserved motif into the RdnI variants from *Prevotella* and *Pseudomonas*. These *Prevotella* and *Pseudomonas* hybrid proteins did not protect against $^{Proteus}$RdnE in the swarm competition assay (*Figure 4H*) or bind to it in the BACTH assay (*Figure 4I*), indicating that the conserved motif is not required for binding but, alone, is insufficient to confer protection. Thus, RdnI-like immunity proteins containing this conserved motif can protect against non-cognate effector proteins if binding has been established.

## RdnE and RdnI proteins from diverse phyla are present in individual human microbiomes

Our findings revealed that immunity proteins such as RdnI could provide a broader protective umbrella for a cell beyond inhibiting the effector proteins of their siblings. If so, one would expect to find evidence of RdnE and RdnI homologs from different phyla in the same environment or microbial community. We tested this hypothesis by analyzing around 500,000 publicly available microbiomes (metagenomes) for the specific *rdnE* and *rdnI* gene sequences examined in *Figure 4* (*Figure 5A*). 2296 human and terrestrial metagenomes contained reads matching with over 90% identity to these *rdnE* sequences (*Figure 5B*). We used this cutoff to ensure that each nucleotide sequence queried in the metagenomes closely matched experimentally characterized reference sequences. As a control, we applied a 70% identity threshold, which would retain related but more divergent sequences. We saw similar patterns with a total ~2% change in the number of genomes per category (*Figure 5—figure supplement 1*). The reads mapped to the expected niche for each organism, underscoring the presence of the genes encoding these specific effector proteins in naturally occurring human-associated microbiomes.

The *rdnE* and *rdnI* genes from various human-associated bacteria occurred concurrently in individual human oral and, to a lesser extent, gut metagenomes. The *rdnE* and *rdnI* genes from *Rothia* and *Prevotella* co-occurred in approximately 5% of the metagenomes analyzed (*Figure 5C*, *Figure 5—figure supplement 2*). Stringent detection parameters were utilized, so the true number could be higher. We then compared the abundance of *rdnI* to *rdnE* reads, since metagenomic coverage (i.e. the number of short reads that map to a gene) approximates the underlying gene's abundance in the sampled community. In most gut samples, *rdnI* recruited more reads than *rdnE*, although there was substantial variance (*Figure 5D*). These data could indicate the presence of orphan *rdnI* genes, which is consistent with published T6SS orphan immunity alleles (*Bosch et al., 2023*; *Kirchberger et al., 2017*; *Koskiniemi et al., 2014*). These metagenomic patterns suggest that a single community can produce multiple RdnE and RdnI proteins from different phyla, thereby providing a potential for them to interact in a host environment.

## Discussion

Using these results as a foundation, we propose an extension to the prevailing model of selective, cognate EI partners (*Jurėnas and Journet, 2021*) to incorporate 'EI skeleton keys'. In this revised model, flexible ('promiscuous') binding between non-cognate effector and immunity proteins enables broader protection in mixed-species communities (*Figure 5E*). Here, we demonstrated that RdnE-RdnI binding is necessary but not sufficient to neutralize RdnE, which differs from many previously described T6SS immunity proteins. We showed that full-length RdnE, containing both its PD-(D/E)XK domain and variable C-terminal region, is a DNA-degrading endonuclease. Likewise, RdnI requires its entire protein to bind and neutralize RdnE, including a newly identified conserved C-terminal motif. Our findings point to a possible two-step mechanism for how the RdnI immunity protein works: the N-terminal variable-sequence domain mediates binding to an effector, while the C-terminal conserved domain contributes to neutralization in a not-yet-determined mechanism (*Figure 5E*). These findings have potential impacts on our understanding of immunity protein evolution, molecular functions, and microbial community structure.

The domain architectures of RdnE and RdnI suggest possible evolutionary trajectories for these EI pairs. While both require the full-length protein, RdnE and RdnI can function with residue changes within the domains and even retain activity in cross-phyla hybrid proteins. Tri1 immunity proteins

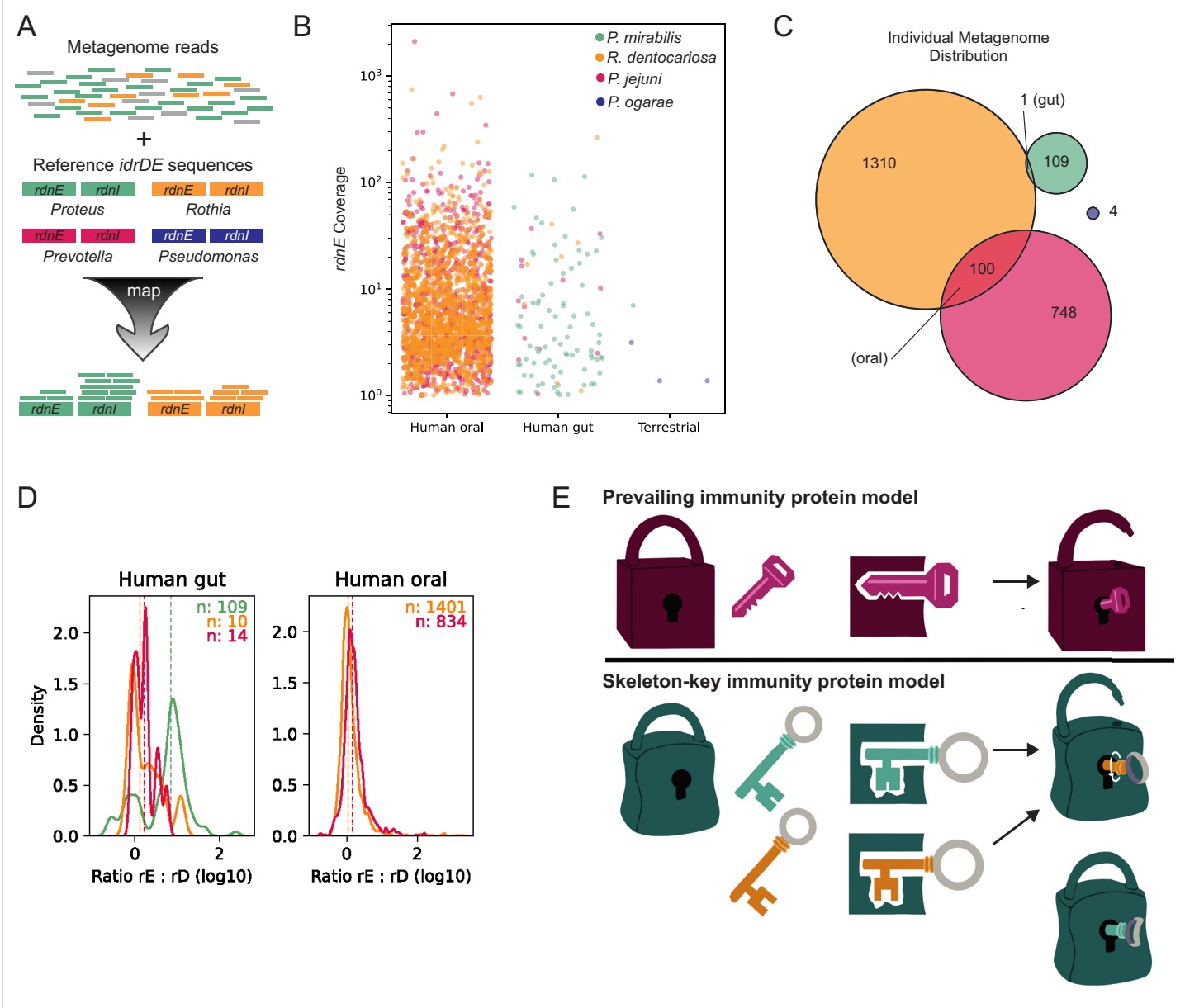

**Figure 5.** The RdnI protein family has the potential for broader protection within oral and gut microbiomes. (**A**) Methodology used to identify *rdnE and rdnI* genes in publicly available metagenomic data. Metagenomes were mapped against sequences with a stringency of 90%. 'Coverage' denotes the average depth of short reads mapping to a gene in a single sample. Colors represent *rdnE* and *rdnI* from *P. mirabilis* (green), *R. dentocariosa* (orange), *P. jejuni* (magenta), or *P. ogarae* (dark blue). (**B**) The experimentally tested *rdnE* gene sequences from different organisms (colors) are found in thousands of human-associated metagenomes. Each dot represents a single sample's coverage of an individual *rdnE* gene, note log$_{10}$-transormed y-axis. Only samples with >1 x coverage are shown. (**C**) Euler diagram showing the number of samples with co-occurring *rdnE* genes from different taxa (colors). (**D**) Kernel density plot of the ratio of *rdnI* to *rdnE* coverage. The ratio of *rdnI* to *rdnE* was defined as log$_{10}$(I/E) where I and E are the mean nucleotide's coverage for *rdnI* and *rdnE*, respectively. The distribution of ratios was summarized as a probability density function (PDF) for each taxon (color) in each environment (subpanel). Here, the y-axis (unitless) reflects the probability of observing a given ratio (x-axis) in that dataset. The colored numbers in the top right of each panel show the number of metagenomes above the detection limit for both *rdnE* and *rdnI* for each taxon. Dashed vertical lines represent the median ratio. (**E**) Skeleton-key model for immunity protein protection. Top, the current prevailing model for T6SS immunity proteins is that protection is defined by necessary and sufficient binding between cognate effectors (locks) and immunity proteins (keys). Bottom, a proposed, expanded model: multiple immunity proteins (skeleton-keys) can bind a single effector due to a flexible (promiscuous) binding site. Protection is a two-step process of binding and then neutralization.

The online version of this article includes the following figure supplement(s) for figure 5:

**Figure supplement 1.** Metagenomic analysis with a 70% stringency revealed similar patterns in RdnE and RdnI localization.

**Figure supplement 2.** RdnE and RdnI sequences are found in metagenomic datasets.

also contain two domains, but these domains are associated with distinct functions. Their conserved region corresponds with the broad-acting enzymatic domain, which is independent of their cognate-specific binding domain (*Ting et al., 2018*), suggesting that conserved enzymatic activity can be maintained alongside potential coevolution necessary for strict cognate pair binding. Therefore, individual domains in EI proteins may evolve independently rather than the entire protein experiencing coevolution. As such, RdnE's PD-(D/E)XK motif and RdnI's conserved motif might be maintained for activity, while the variable domains may diversify in sequence independently. Depending on the selective pressures, the variable regions could reinforce specificity between cognate EI pairs as they coevolve. Additional evolutionary analysis would reveal how the balance between specificity and flexibility evolves in EI pairs, both within domains and across the entire protein.

RdnI's potential two-step mechanism adds to a growing number of ways in which immunity proteins neutralize effector proteins. However, RdnI's neutralization mechanism remains unknown. Many crystalized structures of EI complexes show that immunity proteins can bind and occlude an effector's active site, effectively neutralizing the effector's function (*Benz et al., 2012*; *Hagan et al., 2023*). However, some immunity proteins allosterically inhibit their effector without blocking the active site (*Kleanthous et al., 1999*; *Lu et al., 2014*). Recent studies revealed more neutralization mechanisms. The Tri1 immunity protein has a conserved enzymatic function that neutralizes its effector's activity, allowing for protection against foreign effectors in addition to selective cognate-binding activity (*Ting et al., 2018*), while the Tdi1 immunity protein conformationally disrupts its effector, Tde1's, binding and active sites to prevent DNA nuclease activity (*Bosch et al., 2023*). One will need to experimentally determine the structure of the RdnE-RdnI complex is necessary to define how it neutralizes RdnI and how this molecular mechanism compares to other immunity proteins.

Regardless, our results indicate that RdnI's conserved domain is essential for protective activity and that a combination of seven highly conserved residues mediates that protection. There are several possibilities for how this domain aids neutralization, such as an ion-binding pocket, structural stability, or protein partner binding. Another possibility is that RdnI's conserved region reinforces binding to the effector, aiding non-cognate interactions or co-evolving pairs. For example, while our assays indicate the primary binding domain for RdnI is in the N-terminus, the conserved domain could reinforce an initial, transient binding interaction. Indeed, multiple binding domains have been recorded for TA systems and likely protect the cell during co-evolution (*Grabe et al., 2021*). Binding affinities between effector and immunity proteins are not well-documented; those reported vary. For DNA nuclease colicins, non-cognate interactions have affinities in the nanomolar range whereas cognate interactions have picomolar affinities (*Li et al., 2004*). However, the T6SS-associated EI pair, Tde1 and Tdi1, have similar nanomolar affinities for both cognate and non-cognate orphan EI pairs (*Bosch et al., 2023*). In addition, it is unclear what equilibrium dissociation constant ($K_D$) for EI binding would confer protection in native systems. This $K_D$ may be especially important in the case of highly motile bacteria, such as *P. mirabilis*, that only need to survive long enough to escape. Additionally, interbacterial effectors act within a neighboring cell, which may make determining the native ratios of effector to immunity proteins challenging. It will be interesting to see how binding affinities between other EI pairs compare, both cognate and non-cognate interactions, and how these affinities relate to protection in mixed communities.

When considering the impacts on community structure, the broadened activity of RdnI proteins against RdnE effectors from multiple phyla likely increases bacterial fitness, which is advantageous in dense environments. Our analysis measured protection during a two-dimensional, swarm-structured competition, where RdnI offered a susceptible strain protection against trans-cellular RdnE delivered natively. As such, we can conclude that RdnI production increased individual cells' fitness and modified the community structure; it enabled vulnerable bacteria to inhabit previously restricted spaces. Supporting this experimental data, both gut and oral metagenomes showed evidence of multiple *rdnE-rdnI* pairs within individual samples, particularly between *Rothia* and *Prevotella*. Interestingly, the oral microbiome had roughly equivalent abundance between the effector and immunity genes, which might reflect that bacteria occupy distinct spatiotemporal niches within oral microbiomes, e.g. *R. dentocariosa* is predominantly on tooth surfaces (*Mark Welch et al., 2019*). By contrast, *rdnI* genes had greater abundance compared to *rdnE* in the gut microbiomes, which may reflect the greater diversity in member species and community structures found in the gut (*Donaldson et al., 2016*). Orphan immunity genes are indeed a known phenomenon in T6SS EI literature but are usually

documented through single isolate sequencing, with notable exceptions such as *Ross et al., 2019* and *Bosch et al., 2023*. This community-level assessment affirms the presence of *rdnI* orphan genes on a population scale and points to relatively widespread immunity genes in hundreds or thousands of samples.

Given the ability of immunity genes to protect against non-cognate effectors, the presence of diverse orphan *rdnI* genes hints at the ecological complexity surrounding RdnE and RdnI. This community of immunity proteins is reminiscent of the model for shared immunity proteins within an ecosystem, called a 'hyper-immunity state', which was seen among colicins in wild field mice (*Riley and Wertz, 2002*). In this hyper-immunity state, a set of immunity proteins shared among a community could offer an advantage against pathogens. Invading bacteria would be unable to defend themselves from certain effectors, while the community would be protected as they share the collective immunity proteins. Flexible binding like RdnI could contribute to such a 'hyper-immunity state' to help a bacterial community maintain its niche.

Indeed, bacteria have a diverse set of protective measures to ward off foreign effectors in addition to flexible immunity proteins. Recent work has identified non-specific mechanisms of protection including stress-response, physical barriers, and a stronger offense (*Hersch et al., 2020*). Orphan immunity genes also exist throughout many bacterial genomes and may be a part of this system (*Barretto and Fowler, 2020*), for example, orphan immunity genes offer a fitness advantage in vitro (*Bosch et al., 2023*; *Hagan et al., 2023*) and in mouse microbiomes (*Ross et al., 2019*). Flexible EI pairs are also not limited to secretion systems but are also seen among TA pairs (*Aakre et al., 2015*) and bacteriocins (*Franz et al., 2000*; *Li et al., 2004*). Our data extends the current repertoire of protection mechanisms by adding another tool: a flexible immunity protein collection, where each immunity protein acts as a skeleton key against a wider class of effectors. This flexibility is seen among orphan immunity proteins (*Bosch et al., 2023*; *Hagan et al., 2023*) and for immunity proteins with cognate effectors as in this study. Flexible binding could be a general property of T6SS immunity proteins that could be useful in dense, diverse communities, like human and soil microbiomes, where contact-dependent competition using EI pairs is critical to maintain one's population. Indeed, the physical interactions between, and evolution of, effector and immunity proteins remain a rich area for new explorations.

## Materials and methods
### Bacterial strains and media
All strains are described in *Table 2*. Strains for bacterial two-hybrid assays were transformed the day before. Overnight cultures were grown aerobically at 37 °C in LB (Lennox) broth (*Belas et al., 1991*). *E. coli* strains were plated on LB (Lennox) agar surfaces (1.5% Bacto agar) and *P. mirabilis* strains were plated on LSW⁻ agar (*Belas et al., 1991*) for single-colony growth or 25 mL CM55 media (Thermo Fisher Scientific Cat# CM0055B) for swarms. When necessary 35 µg/mL kanamycin or 100 µg/mL carbenicillin was included in the media.

### Plasmid construction
Plasmids were constructed according to *Table 3*. Primers and gBlocks were ordered from Integrated DNA Technologies (IDT), Coralville, IA. P$_{idrA}$-RdnE was constructed using Polymerase Chain Reaction (PCR) to amplify the last 416 bp of the *idrD* gene from BB2000 and clone it into the SacI and AgeI sites of the pBBR1-NheI vector, resulting in plasmid pAS1054. RdnE is the final 138 amino acids of IdrD (out of its total of 1581). P$_{idrA}$-*rdnE-rdnI* was constructed by PCR amplifying the last 416 bp of the *idrD* gene through the end of the *rdnI* gene from BB2000, resulting in the plasmid pAS1059. The gBlock and primer sequences are archived on an OSF website (*https://osf.io/scb7z/*).

We used several standard protocols for vector construction. Seamless ligation cloning extract (SliCE) was adapted from *Zhang et al., 2014*. Restriction-digest reactions were based on manufacturer's protocols. Overlap extension (SOE) PCR Amplification was adapted from *Heckman and Pease, 2007*. Plasmids were transformed into OmniMax *E. coli* and confirmed using Sanger Sequencing (UC Berkeley DNA Sequencing Facility and Genewiz, South Plainfield NJ).

**Table 2.** List of strains used in this study.

| Strain | Strain Name | Description | Reference or Source |
|---|---|---|---|
| *P. mirabilis* BB2000 *idrD*\*+pDS0062 | DS349 | BB2000 *idrD*::Tn5 (Cm$^R$) producing GFPmut2 under the aTc-inducible promoter | This study |
| *P. mirabilis* BB2000 *idrD*\*+pDS0002 | DS104 | BB2000 *idrD*::Tn5 (Cm$^R$) producing RdnE under the aTc-inducible promoter | This study |
| *P. mirabilis* BB2000 *idrD*\*+pDS0058 | DS344 | BB2000 *idrD*::Tn5 (Cm$^R$) producing RdnE$_{D39A}$ under the aTc-inducible promoter | This study |
| *P. mirabilis* BB2000 *idrD*\*+pDS0059 | DS345 | BB2000 *idrD*::Tn5 (Cm$^R$) producing RdnE$_{E53A}$ under the aTc-inducible promoter | This study |
| *P. mirabilis* BB2000 *idrD*\*+pDS0060 | DS346 | BB2000 *idrD*::Tn5 (Cm$^R$) producing RdnE$_{K55A}$ under the aTc-inducible promoter | This study |
| *P. mirabilis* BB2000 *idrD*\*+pDS0061 | DS347 | BB2000 *idrD*::Tn5 (Cm$^R$) producing RdnE$_{D39A}$ E53A K55A under the aTc-inducible promoter | This study |
| *E. coli* MG1655 +pBBR1-NheI | DS068 | MG1655 carrying empty vector | This study |
| *E. coli* MG1655 +pDS0002 | DS151 | MG1655 producing RdnE under the aTc-inducible promoter | This study |
| *E. coli* MG1655 +pDS0058 | DS336 | MG1655 producing RdnE$_{D39A}$ under the aTc-inducible promoter | This study |
| *E. coli* MG1655 +pDS0059 | DS337 | MG1655 producing RdnE$_{E53A}$ under the aTc-inducible promoter | This study |
| *E. coli* MG1655 +pDS0060 | DS338 | MG1655 producing RdnE$_{K55A}$ under the aTc-inducible promoter | This study |
| *E. coli* MG1655 +pDS0061 | DS339 | MG1655 producing RdnE$_{D39A\ E53A\ K55A}$ under the aTc-inducible promoter | This study |
| *P. mirabilis* BB2000 *idrD*\*+pDS0003 | DS092 | BB2000 *idrD*::Tn5 (Cm$^R$) co-producing RdnE followed by RdnI under the aTc-inducible promoter | This study |
| *E. coli* MG1655 +pDS0003 | DS170 | MG1655 co-producing RdnE followed by RdnI under the aTc-inducible promoter | This study |
| *P. mirabilis* BB2000 +pBBR1-NheI | ANS1127 | BB2000 carrying empty vector | *Wenren et al., 2013* |

*Table 2 continued on next page*

*Table 2 continued*

| Strain | Strain Name | Description | Reference or Source |
|---|---|---|---|
| *P. mirabilis* ATCC29906 +pBBR1-NheI | ANS1280 | ATCC29906 carrying empty vector | This study |
| *P. mirabilis* ATCC29906 +pAK043 | AK0132 | ATCC29906 producing RdnI with a C-terminal Strep-II tag with the *fla* promoter | This study |
| *P. mirabilis* ATCC29906 +pLMW04-gfp | AK387 | ATCC29906 producing GFPmut2 with the *fla* promoter | This study |
| *E. coli* MG1655 +pDS0048 | DS248 | MG1655 producing RdnE$_{D39A}$ with a C-terminal FLAG tag under an aTc-inducible promoter | This study |
| *E. coli* BL21(pLysS)DE3 +pAK023 | AK024 | BL21(pLysS)DE3 producing RdnI with a C-terminal Strep-II tag under the T7 promoter | This study |
| *P. mirabilis* ATCC29906 +pAK044 | AK0135 | ATCC29906 producing the $^{Rothia}$RdnI with a C-terminal Strep-II tag under the *fla* promoter | This study |
| *P. mirabilis* ATCC29906 +pAK045 | AK0138 | ATCC29906 producing the $^{Prevotella}$RdnI with a C-terminal Strep-II tag under the *fla* promoter | This study |
| *P. mirabilis* ATCC29906 +pAK046 | AK0141 | ATCC29906 producing the $^{Pseudomonas}$RdnI with a C-terminal Strep-II tag under the *fla* promoter | This study |
| *E. coli* BL21(pLysS)DE3 +pAK058 | AK261 | BL21(pLysS)DE3 producing $^{Rothia}$RdnI with a C-terminal Strep-II tag under the T7 promoter | This study |
| *E. coli* BL21(pLysS)DE3 +pAK059 | AK262 | BL21(pLysS)DE3 producing $^{Prevotella}$RdnI with a C-terminal Strep-II tag under the T7 promoter | This study |
| *E. coli* BL21(pLysS)DE3 +pAK060 | AK263 | BL21(pLysS)DE3 producing $^{Pseudomonas}$RdnI with a C-terminal Strep-II tag under the T7 promoter | This study |
| *P. mirabilis* ATCC29906 +pAK063 | AK318 | ATCC29906 producing the recoded $^{Proteus}$RdnI sequence with a C-terminal Strep-II tag under the *fla* promoter | This study |
| *P. mirabilis* ATCC29906 +pAK065 | AK320 | ATCC29906 producing the $^{Rothia}$RdnI sequence (aa195-271) inserted between aa192-266 in the recoded $^{Proteus}$RdnI with a C-terminal Strep-II tag under the *fla* promoter | This study |
| *P. mirabilis* ATCC29906 +pAK066 | AK321 | ATCC29906 producing the $^{Prevotella}$RdnI sequence (aa170-245) inserted between aa192-266 in the recoded $^{Proteus}$RdnI with a C-terminal Strep-II tag under the *fla* promoter | This study |

*Table 2 continued on next page*

Table 2 continued

| Strain | Strain Name | Description | Reference or Source |
|---|---|---|---|
| P. mirabilis ATCC29906 +pAK067 | AK322 | ATCC29906 producing the [Pseudomonas]RdnI sequence (aa181-255) inserted between aa192-266 in the recoded [Proteus]RdnI with a C-terminal Strep-II tag under the *fla* promoter | This study |
| P. mirabilis ATCC29906 +pAK086 | AK381 | ATCC29906 producing the recoded [Proteus]RdnI sequence (aa192-266) inserted between aa170-245 in the [Prevotella]RdnI with a C-terminal Strep-II tag under the *fla* promoter | This study |
| P. mirabilis ATCC29906 +pAK087 | AK382 | ATCC29906 producing the recoded [Proteus]RdnI sequence (aa192-266) inserted between aa181-255 in the [Pseudomonas]RdnI with a C-terminal Strep-II tag under the *fla* promoter | This study |
| P. mirabilis ATCC29906 +pAK064 | AK319 | ATCC producing the recoded [Proteus]RdnI sequence with seven alanine mutations (Y197A, H221A, S235A, P244A, E246A, R254A, K258A) and a C-terminal Strep-II tag under the *fla* promoter | This study |
| OneShot OmniMax 2 T1R Competent Cells | | E. coli strain for cloning | Thermo Fisher Scientific, Waltham, MA |
| MFDpir | | Mu-free E. coli mating strain to introduce plasmids into P. mirabilis | *Ferrières et al., 2010* |

**Table 3.** Plasmids used in this study.

| Plasmid Name | Description | Cloning Method or Source |
|---|---|---|
| pBBR1-NheI | empty vector with pBBR1 origin. | *Gibbs et al., 2008* |
| pLMW04-gfp | GFPmut2 with a constitutive *fla* promoter, (pBBR1 origin, Kan resistance). | *Wenren et al., 2013* |
| pDS0002 | *rdnE* with the anhydrotetracycline (aTc)-inducible promoter, $P_{tet}$, (pBBR1 origin, Kan resistance) | gDS0005 was recombined into amplified pAS1054 by SliCE |
| pDS0062 | *gfpmut2* with the aTc-inducible promoter, (pBBR1 origin, Kan resistance) | restriction digest using amplified *gfpmut2* and pDS0002 |
| pDS0048 | *rdnE*$_{D39A}$-FLAG with the aTc-inducible promoter, (pBBR1 origin, Kan resistance) | gDS0025 was recombined into pDS0002 using restriction digest |
| pDS0058 | *rdnE*$_{D39A}$ with the aTc-inducible promoter, (pBBR1 origin, Kan resistance) | pDS0048 was recombined into pDS0002 using restriction digest |
| pDS0059 | *rdnE*$_{E53A}$ with the aTc-inducible promoter, (pBBR1 origin, Kan resistance) | gDS0026 was recombined into pDS0002 using restriction digest |
| pDS0060 | *rdnE*$_{K55A}$ with the aTc-inducible promoter, (pBBR1 origin, Kan resistance) | gDS0027 was recombined into pDS0002 using restriction digest |
| pDS0061 | *rdnE*$_{D39A, E53A, K55A}$ with the aTc-inducible promoter, (pBBR1 origin, Kan resistance) | gDS0028 was recombined into pDS0002 using restriction digest |
| pDS0034 | *rdnE*-FLAG with the aTc-inducible promoter, (pBBR1 origin, Kan resistance) | gDS0023 was recombined into amplified pDS0002 using SliCE |
| pDS0003 | *rdnE-rdnI* with the aTc-inducible promoter, (pBBR1 origin, Kan resistance) | Amplified *rdnE-rdnI* from pAS1059 was recombined into pDS0002 using SOE PCR |
| pAK023 | *rdnI*-StrepII with the T7 promoter, (pUC, Amp resistance) | gAK001 and pDS0003 were recombined into pET17b vector using SOE PCR |
| pAK043 | *rdnI*-StrepII with the *fla* promoter, (pBBR1 origin, Kan resistance) | *rdnI*-StrepII amplified from pAK023 was recombined into pLMW04 using restriction digest |
| pAK070 | T25-linker-*rdnE*$_{D39A}$-FLAG with the IPTG-inducible *lac* promoter, (p15A, Kan resistance) | *rdnE*$_{D39A}$-FLAG tag amplified from pDS0048 was recombined into pKT25 using restriction digest |

*Table 3 continued on next page*

*Table 3 continued*

| Plasmid Name | Description | Cloning Method or Source |
|---|---|---|
| pAK071 | T25-linker-*rdnI* with the *lac* promoter, (p15A, Kan resistance) | *rdnI*-Strep-II amplified from pAK043 was recombined into pKT25 using restriction digest |
| pAK074 | T18-linker-*rdnE*$_{D39A}$-FLAG with the *lac* promoter, (Col E1 origin, Amp resistance) | *rdnE*$_{D39A}$-FLAG tag amplified from pDS0048 was recombined into pUT18C using restriction digest |
| pAK075 | T18-linker-*rdnI*-StrepII with the *lac* promoter, (Col E1 origin, Amp resistance) | *rdnI*-Strep-II tag amplified from pAK043 was recombined into pUT18C using restriction digest |
| pAK076 | T25-*gfpmut2* with the *lac* promoter, (p15A origin, Kan resistance) | Amplified *gfpmut2* was recombined into pKT25 using restriction digest |
| pAK077 | T18-*gfpmut2* with the *lac* promoter, (Col E1 origin, Amp resistance) | Amplified *gfpmut2* was recombined into pUT18C using restriction digest |
| pAK044 | $^{Rothia}$*rdnI*-StrepII with the *fla* promoter, (pBBR1 origin, Kan resistance) | gAK003 was recombined into pAK043 using restriction digest |
| pAK045 | $^{Prevotella}$*rdnI*-StrepII with the *fla* promoter, (pBBR1 origin, Kan resistance) | gAK004 was recombined into pAK043 using restriction digest |
| pAK046 | $^{Pseudomonas}$*rdnI*-StrepII with the *fla* promoter, (pBBR1 origin, Kan resistance) | gAK005 was recombined into pAK043 using restriction digest |
| pAK079 | T18-$^{Rothia}$*rdnI*-StrepII with the *lac* promoter, (pUC, Amp resistance) | gAK003 was recombined into pUT18C using restriction digest |
| pAK081 | T18-$^{Prevotella}$*rdnI*-StrepII with the *lac* promoter, (pUC, Amp resistance) | gAK004 was recombined into pUT18C using restriction digest |
| pAK083 | T18-$^{Pseudomonas}$*rdnI*-StrepII with the *lac* promoter, (pUC, Amp resistance) | gAK005 was recombined into pUT18C using restriction digest |
| pAK058 | $^{Rothia}$*rdnI*-StrepII with the T7 promoter, (pUC, Amp resistance) | gAK003 was recombined into pAK023 using restriction digest |
| pAK059 | $^{Prevotella}$*rdnI*-StrepII with the T7 promoter, (pUC, Amp resistance) | gAK004 was recombined into pAK023 using restriction digest |

*Table 3 continued on next page*

*Table 3 continued*

| Plasmid Name | Description | Cloning Method or Source |
|---|---|---|
| pAK060 | $^{Pseudomonas}$rdnI-StrepII with the T7 promoter, (pUC, Amp resistance) | gAK005 was recombined into pAK023 using restriction digest |
| pAK063 | Sequence optimized rdnI-StrepII with the *fla* promoter, (pBBR1 origin, Kan resistance) | gAK024 was recombined into pAK043 using restriction digest |
| pAK065 | $^{Pm/Rd}$rdnI-StrepII (*Proteus rdnI* with *Rothia* conserved motif insert) with the *fla* promoter, (pBBR1 origin, Kan resistance) | gAK026 was recombined into pAK043 using restriction digest |
| pAK066 | $^{Pm/Pj}$rdnI-StrepII (*Proteus rdnI* with *Prevotella* conserved motif insert) with the *fla* promoter, (pBBR1 origin, Kan resistance) | gAK028 was recombined into pAK043 using restriction digest |
| pAK067 | $^{Pm/Po}$rdnI-StrepII (*Proteus rdnI* with *Pseudomonas* conserved motif insert) with the *fla* promoter, (pBBR1 origin, Kan resistance) | gAK027 was recombined into pAK043 using restriction digest |
| pAK086 | $^{Pj/Pm}$rdnI-StrepII (*Prevotella rdnI* with *Proteus* conserved motif insert) with the *fla* promoter, (pBBR1 origin, Kan resistance) | gAK029 was recombined into pAK043 using restriction digest |
| pAK087 | $^{Pf/Pm}$rdnI-StrepII (*Pseudomonas rdnI* with *Proteus* conserved motif insert) with the *fla* promoter, (pBBR1 origin, Kan resistance) | gAK030 was recombined into pAK043 using restriction digest |
| pAK092 | T18-linker-Sequence optimized $^{Proteus}$rdnI with a C-terminal StrepII with the *lac* promoter (Col E1 origin, Amp resistance) | gAK024 was recombined into pAK075 using restriction digest |
| pAK093 | T18-linker-Sequence optimized $^{Pm/Ra}$rdnI (*Proteus rdnI* with *Rothia* conserved motif insert) with a C-terminal StrepII with the *lac* promoter (Col E1 origin, Amp resistance) | gAK026 was recombined into pAK075 using restriction digest |
| pAK094 | T18-linker-Sequence optimized $^{Pm/Pj}$rdnI (*Proteus rdnI* with *Prevotella* conserved motif insert) with the *lac* promoter (Col E1 origin, Amp resistance) | gAK028 was recombined into pAK075 using restriction digest |
| pAK095 | T18-linker-Sequence optimized $^{Pm/Pf}$rdnI (*Proteus rdnI* with *Pseudomonas* conserved motif insert) with the *lac* promoter (Col E1 origin, Amp resistance) | gAK027 was recombined into pAK075 using restriction digest |
| pAK096 | T18-linker- $^{Pj/Pm}$rdnI (*Prevotella rdnI* with *Proteus* conserved motif insert) with the *lac* promoter (Col E1 origin, Amp resistance) | gAK029 was recombined into pAK075 using restriction digest |
| pAK097 | T18-linker-$^{Pf/Pm}$rdnI (*Pseudomonas rdnI* with *Proteus* conserved motif insert) with a C-terminal StrepII with the *lac* promoter (Col E1 origin, Amp resistance) | gAK030 was recombined into pAK075 using restriction digest |
| pAK064 | Sequence optimized rdnI$_{7mut}$-StrepII with the *fla* promoter, (pBBR1 origin, Kan resistance) | gAK025 was recombined into pAK043 using restriction digest |

*Table 3 continued on next page*

*Table 3 continued*

| Plasmid Name | Description | Cloning Method or Source |
|---|---|---|
| pAK085 | T18-linker-Sequence optimized *rdnI*$_{7mut}$-StrepII with the *lac* promoter (Col E1 origin, Amp resistance) | gAK025 was recombined into pAK075 using restriction digest |

## In vitro DNase assay

RdnE proteins were produced using the New England Biolabs PURExpress In Vitro Protein Synthesis Kit (New England BioLabs Inc, Ipswich MA). Template DNA contained the *rdnE* gene and required elements specified by the PURExpress kit. We adapted this protocol from prior in vitro DNA-degradation assays (**Hughes and Cidlowski, 1997**). Reactions were performed with 250 ng of template DNA and incubated at 37 °C for 2 hr (no template DNA added to negative control reaction). The protein amount was determined using an anti-FLAG western blot with a known gradient of FLAG-BAP (2.5, 5, 10, and 20 ng). Synthesized protein (2.5, 5, and 10 ng) was added to 0.5 µg of lambda DNA (methylated and unmethylated), 5 µL of New England Biolabs Buffer 3.1, and up to a final volume of 25 µL. For plasmid DNase assays, 10 ng of synthesized protein was added to 250 ng of circular or linear plasmid DNA (pids$_{BB}$ [**Gibbs et al., 2008**]). This reaction was incubated for 1 hr at 37 °C, then Proteinase K (New England Biolabs Inc, Ipswich MA) was added and incubated for an additional 15 min at 37 °C. The reaction was then run on a 1% agarose gel for analysis.

## *E. coli* liquid growth and viability assays

Overnight cultures were grown at 37 °C in a shaking incubator in LB broth with appropriate antibiotics. Cultures were normalized to an optical density at 595 nm (OD$_{595}$) of 1 and diluted 1:100 into LB broth with 35 µg/mL kanamycin, with and without 200 nM anhydrotetracycline (aTc). Samples were analyzed for OD$_{595}$ every thirty minutes for 16 hr in a 96-well plate using a TECAN. Other samples were incubated at 37 °C for 6 hr while rocking. At indicated time points, 100 µL of sample was removed, diluted, and then plated on fresh LB agar plates to measure colony- forming units per mL (CFU) after overnight growth at 37 °C using standard protocols.

## Microscopy

We performed microscopy on *P. mirabilis* strain *idrD*::Tn5 (Cm$^R$) (also called, *idrD*\*), which has a transposon insertion to disrupt *rdnE* and *rdnI* expression (**Wenren et al., 2013**), carrying either vector pBBR1-NheI or pDS0002 (producing RdnE) and on *E. coli* carrying either pBBR1-NheI, pDS0002, or pDS0048 (producing RdnE$_{D39A}$). *P. mirabilis* cells were normalized to OD$_{595}$ of 0.1 after overnight growth in LB broth supplemented with kanamycin. Cells were inoculated onto CM55 swarm pads containing 10 µg/mL DAPI and 10 nM aTc and grown in humidified chambers at 37 °C. Images were taken at five and six hours after growth. From overnight cultures, *E. coli* cells were grown in LB broth plus kanamycin until mid-logarithmic phase and then mounted directly onto glass slides. Glass coverslips were sealed with nail polish. For all microscopy, we captured phase contrast and DAPI (150ms exposure) images using a Leica DM5500B microscope (Leica Microsystems, Buffalo Grove IL) and CoolSnap HQ CCD camera (Photometrics, Tucson AZ) cooled to –20 °C. MetaMorph version 7.8.0.0 (Molecular Devices, Sunnyvale CA) was used for image acquisition.

## Sequence-optimized RdnI

The *P. mirabilis rdnI-StrepII* sequence was difficult to genetically engineer due to its low GC% content (23%). As such, we engineered the sequence to have a higher GC%, called 'Sequence optimized (SO) $^{Proteus}$RdnI-StrepII' without changes to its amino acid sequence. The change to the nucleotide sequence did not affect the construct's ability to offer protection to a vulnerable strain (**Figure 4B**).

## Swarm competition assay

The swarm competition (territoriality) assay was adapted from **Wenren et al., 2013**. 5 mL cultures were grown in LB broth with appropriate antibiotics overnight in a 37 °C rocking incubator. Overnight

cultures were normalized to an OD$_{595}$ of 1. For the competition samples, the strains were mixed 1:1. 2 µL of each sample was inoculated onto CM55 agar with the appropriate antibiotic. Plates were incubated at 37 °C for 22 hr and then photographed and assessed for boundary formation. All RdnI-producing strains contained a low-copy vector with the *rdnI* gene under the *fla* constitutive promoter; see *Figure 4—figure supplement 2* for relative protein production.

## BACTH assay

The vectors are described in *Battesti and Bouveret, 2012* with an added linker region between the T25 or T18 fragments and multiple cloning sites. BTH101 cultures were grown at 30 °C overnight in LB broth with kanamycin and carbenicillin. 10 µL of the overnight culture were inoculated onto LB agar with kanamycin, carbenicillin, 1 mM IPTG, and 0 or 40 µg/mL of X-gal (Thermo Fisher, Waltham MA), and grown at 30 °C for 24 hr. Color was amplified by an additional 24 hr at 4 °C, and then samples were imaged.

## FLAG co-immunoprecipitation assays

The protocol was adapted from *Cardarelli et al., 2015*. *E. coli* cells were harvested from LB broth, grown for either 3 hr after induction with 200 nM aTc at 37 °C or 16–20 hr at 16 °C after induction with 1 mM IPTG. Cells were spun down into pellets using centrifugation and then flash frozen in liquid nitrogen. RdnE-containing samples were lysed in 50 mM Tris pH 7.4, 150 mM NaCl, and 1 x Protease Inhibitor Cocktail (Selleck Chemicals LLC, Houston TX), via bead bashing for 20 min at 4 °C. RdnI-containing samples were lysed in 100 mM Tris-HCl pH 8, 180 mM NaCl, and 1 x Protease Inhibitor Cocktail via 10x10 s sonication pulses. The soluble fraction for both samples was obtained after centrifugation at 15,000 rpm for 15 min. FLAG epitope-containing samples were incubated with prepared resin for 2 hr at 4 °C. The resin was then washed twice (50 mM Tris pH 7.4, 150 mM NaCl, and 1% Tween-20), incubated with approximately 1 mL of the soluble fraction of the RdnI-StrepII-containing samples for another 2 hr, and washed thrice more. The protein was finally eluted with 50 µL of 300 ng/µL 3 x FLAG peptide (Sigma-Aldrich, St. Louis, MO) for 45 min at 4 °C. Sample buffer (63 mM Tris pH 6.8, 2% Sodium Dodecyl Sulfate, 10% glycerol, 5% 2-Mercaptoethanol) was added to samples, boiled at 95 °C for 10 min, and frozen at –80 °C.

## *P. mirabilis* swarm cell protein expression

The protocol was adapted from *Cardarelli et al., 2015*. *P. mirabilis* cells were harvested from CM55 swarm plates, grown overnight at 37 °C. Cells were washed twice in LB broth, spun down by centrifugation, and flash frozen in liquid nitrogen. Cells were lysed in 100 mM Tris-HCl pH 8, 180 mM NaCl, and 1 x Protease Inhibitor Cocktail via 10 x via bead bashing for 20 min at 4 °C. The whole cell extract was obtained after centrifugation at 6000 × *g* for 15 min. The soluble fraction for both samples was obtained after subsequent centrifugation at 15,000 rpm for 15 min. Sample buffer (63 mM Tris pH 6.8, 2% Sodium Dodecyl Sulfate, 10% glycerol, 5% 2-Mercaptoethanol) was added to samples, boiled at 95 °C for 10 min, and then immediately used for SDS-PAGE and western blotting.

## SDS-PAGE and western blotting

The protocol was adapted from *Cardarelli et al., 2015*. Protein samples were separated by gel electrophoresis using 13% Tris-Tricine polyacrylamide gels and either transferred to a 0.45 µm nitrocellulose membrane (Bio-Rad Laboratories, Hercules CA) or stained with Coomassie blue (Bio-Rad Laboratories, Hercules CA). Western blot membranes were probed with primary antibody, either 1:4000 rabbit anti-FLAG (Sigma-Aldrich Cat# F3165, RRID:AB_259529) or 1:4000 or 1:2000 mouse anti-StrepII (GenScript Cat# A01732, RRID:AB_2622218) for 1 hr at room temperature or overnight at 4 °C and with secondary antibody either 1:5000 goat anti-rabbit or anti-mouse respectively conjugated to horseradish peroxidase (HRP) (SeraCare KPL Cat# 5220–0395, RRID:AB_3698113 and SeraCare KPL Cat# 074–1506, RRID:AB_2721169, respectively) for 30 min for co-IPs or 1 hour for *P. mirabilis* protein expression at room temperature. Samples were finally visualized using Immun-Star HRP substrate kit (Bio-Rad Laboratories, Hercules, CA) and the ChemiDoc XRS system (Bio-Rad Laboratories, Hercules, CA). TIFF files were analyzed on Fiji (ImageJ, Madison, WI).

## Bioinformatics search for RdnE and RdnI homologs

A BLAST (*Altschul et al., 1990*) search of the *P. mirabilis* RdnE protein sequence revealed seven RdnE homologs from a variety of species. The downstream genes of these RdnE homologs were identified

using the DOE Joint Genome Institute (JGI) Integrated Microbial Genomes and Microbiomes (IMG/M) (*Chen et al., 2021*; *Mukherjee et al., 2021*). The seven RdnE and RdnI amino acid sequences were separately aligned with MUSCLE using Jalview (*Edgar, 2004*; *Waterhouse et al., 2009*). These alignments were then used as seeds for a second homology search using HMMER search (HmmerWeb version 2.41.2; *Finn et al., 2015*; *Finn et al., 2011*) and the Ensembl Database (*Cunningham et al., 2022*). The two data sets were then compared for genomes that contained both *rdnE* and *rdnI* genes next to one another within their respective genomes. Any EI pairs that contained disrupted PD-(D/E) XK motifs within their RdnE sequence were removed.

## Gene neighborhood and primary conservation analyses

Gene neighborhoods were obtained using JGI's IMG/M Neighborhood viewer and then redrawn using Adobe Illustrator (Adobe Inc, 2022). Locations of predicted functions are approximate primarily based on the Pfam domain calling by IMG/M. The gene neighborhoods, relevance, and niche were also accessed from IMG/M.

The final 21 RdnE and RdnI sequences were aligned with MUSCLE using Jalview (*Edgar, 2004*; *Waterhouse et al., 2009*). The conserved residues were identified using Jalview, and the cartoons were created using Adobe Illustrator (Adobe Inc, 2022). The sequence logo for the RdnI conserved motif was generated with WebLogo (*Gabler et al., 2020*) and constrained to only visualize the conserved motif. Trees for unrooted maximum likelihood trees of the RdnE and RdnI were created with RaxML (*Kozlov et al., 2019*). The phylogenetic tree is based on NCBI taxonomy. A tanglegram (*Scornavacca et al., 2011*) was made from the RdnE and RdnI protein families from the 21 sequences.

## Secondary and tertiary structure predictions

Secondary structure predictions of the MUSCLE aligned sequences were determined with Ali2D from the MPI Bioinformatics toolkit (*Edgar, 2004*; *Gabler et al., 2020*; *Zimmermann et al., 2018*). The resulting predictions were made into cartoons manually using Adobe Illustrator (Adobe Inc, 2022). Tertiary structure predictions were done with AlphaFold2 (*Jumper et al., 2021*) using Mmseqs2 on Google Colab (*Mirdita et al., 2022*). Query protein sequences were inputted into the program and then run, producing 5 models ranked 1–5. Rank 1 models are shown. pIDDT scores indicate confidence levels for each amino acid position. Structures were analyzed in PyMOL (The PyMOL Molecular Graphics System, Version 2.2.3 Schrödinger, LLC.). The pIDDT graphs are in the supplemental data.

## Metagenomic analyses

A sourmash-based approach was used to screen approximately 500,000 public metagenomes stored on NCBI's SRA (https://github.com/sourmash-bio/2022-search-sra-with-mastiff; *sourmash, 2023*) for the presence of the 10 genomes shown in *Figure 3A*. Hits with a containment score greater than 0.2 were downloaded for further analysis, representing 9137 metagenomes. Each metagenome was then mapped with bbmap (*Bushnell, 2014*) against a reference database the 10 *rdnE* and *rdnI* gene sequence pairs with a stringency of 90% (minid = 0.9), along with quality filtering (trim1=20, minaveragequality = 10). 70% stringency was also included and resulted in similar results (*Figure 5—figure supplement 1*). After mapping, metagenomes were retained if they had (1) a mean coverage greater than 2 X, (2) at least one base covered greater than 5 X, and (3) more than half of the bases on reference *rdnE-rdnI* sequence receiving coverage. Coverage of other domains, if any, upstream of the C-terminal domain was not considered for subsequent analysis. 2857 metagenomes met these criteria, of which 2296 contained *P. mirabilis*, *R. dentocariosa*, *P. jejuni*, or *P. ogarae* sequences and could be confidently assigned to samples obtained from the human gut or oral microbiome or from terrestrial sources. Gene-level coverage in a sample was then summarized as each gene's average nucleotide coverage. The ratio of *rdnI* to *rdnE* coverage was then calculated for each sample and log10-transformed, and the distribution of ratios was summarized with Python's seaborn kdeplot using a bandwidth of 0.4 (*Waskom, 2021*).

## Materials availability statement

The sequence files and associated data, including sequence datasets and protein modeling files, are archived on an OSF website (https://osf.io/scb7z/) and were made publicly available upon publication. Primer and gBlock sequences are available at the OSF link above. Also included are any newly

generated custom and reused computer code. All plasmids and strains are available by contacting the corresponding author and will be shipped in accordance with University of California, Berkeley and relevant authorities expediently upon written request.

All biological experiments were performed a minimum of three independent times, each time with independent isolates. The data describes biological replicates. The authors have no conflicts of interest to report. No human or animal subjects were used in this study.

## Acknowledgements

We thank Caroline Boyd, Niels Bradshaw, Emma Keteku, Alecia Septer, Nora Sullivan, Adnan Syed, and Larissa Wenren for contributing experimental materials to this project. Rachelle Gaudet, Colleen Cavanaugh, and members of the Gibbs Lab provided valuable advice on the manuscript. The David and Lucile Packard Foundation, the George W Merck Fund, the National Institutes of Health (Training Grant number T32GM135143), Harvard University, and the University of California, Berkeley, and funded this research. AK, DS, DU, and KAG designed and performed research as well as analyzed data. AK, DS, DU, and KAG wrote the paper. We have no competing interests to declare.

## Additional information

### Competing interests

Karine A Gibbs: Reviewing editor, eLife. The other authors declare that no competing interests exist.

### Funding

| Funder | Grant reference number | Author |
| --- | --- | --- |
| The David and Lucile Packard Foundation | | Karine A Gibbs |
| The George W. Merck Fund | | Karine A Gibbs |
| National Institutes of Health | T32GM135143 | Abigail Knecht Denise Sirias |
| Harvard University | | Karine A Gibbs |
| University of California Berkeley | | Karine A Gibbs |

The funders had no role in study design, data collection and interpretation, or the decision to submit the work for publication.

### Author contributions

Abigail Knecht, Denise Sirias, Conceptualization, Data curation, Formal analysis, Validation, Investigation, Visualization, Methodology, Writing – original draft, Writing – review and editing; Daniel R Utter, Data curation, Formal analysis, Validation, Investigation, Visualization, Methodology, Writing – original draft, Writing – review and editing; Karine A Gibbs, Conceptualization, Resources, Supervision, Funding acquisition, Validation, Writing – original draft, Project administration, Writing – review and editing

### Author ORCIDs

Abigail Knecht ⓘ https://orcid.org/0000-0003-1084-1623
Daniel R Utter ⓘ https://orcid.org/0000-0003-3322-7108
Karine A Gibbs ⓘ https://orcid.org/0000-0002-1246-6401

Reviewer #3 (Public Review): https://doi.org/10.7554/eLife.90607.3.sa1
Reviewer #4 (Public Review): https://doi.org/10.7554/eLife.90607.3.sa2
Reviewer #5 (Public Review): https://doi.org/10.7554/eLife.90607.3.sa3
Author response https://doi.org/10.7554/eLife.90607.3.sa4

## Additional files

### Supplementary files
MDAR checklist

### Data availability
The sequence files and associated data are stored on a public OSF website (https://osf.io/scb7z/). All data generated or analyzed during this study are included in the manuscript and supporting files.

The following dataset was generated:

| Author(s) | Year | Dataset title | Dataset URL | Database and Identifier |
|---|---|---|---|---|
| Gibbs KA, Knecht A, Utter D, Sirias D, Utter DR | 2025 | Non-cognate immunity proteins provide broader defenses against interbacterial effectors in microbial communities | https://osf.io/scb7z/ | Open Science Framework, scb7z |

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
