## [Editor Report · eLife assessment]

This study provides **valuable** insights into the specificity and promiscuity of toxic effector and immunity protein pairs. While the work is improved over a previous version, there are still some questions regarding the methodology used to draw certain conclusions, rendering the study somewhat **incomplete**. Nevertheless, this work will likely be of interest to microbiologists and biochemists working with toxin-antitoxin systems and effector-immunity proteins.

---

## [Referee Report · Reviewer #3 (Public Review)]

Summary:

The authors discovered that the RdnE effector possesses DNase activity, and in competition, P. mirabilis having RdnE outcompetes the null strain. Additionally, they presented evidence that the RdnI immunity protein binds to RdnE, suppressing its toxicity. Interestingly, the authors demonstrated that the RdnI homolog from a different phylum (i.e., Actinomycetota) provides cross-species protection against RdnE injected from P. mirabilis, despite the limited identity between the immunity sequences. Finally, using metagenomic data from human-associated microbiomes, the authors provided bioinformatic evidence that the rdnE/rdnI gene pair is widespread and present in individual microbiomes. Overall, the discovery of broad protection by non-cognate immunity is intriguing, although not necessarily surprising in retrospect, considering the prolonged period during which Earth was a microbial battlefield/paradise.

Strengths:

The authors presented a strong rationale in the manuscript and characterized the molecular mechanism of the RdnE effector both in vitro and in the heterologous expression model. The utilization of the bacterial two-hybrid system, along with the competition assays, to study the protective action of RdnI immunity is informative. Furthermore, the authors conducted bioinformatic analyses throughout the manuscript, examining the primary sequence, predicted structural, and metagenomic levels, which significantly underscore the significance and importance of the EI pair.

Weaknesses:

(1) The interaction between RdnI and RdnE appears to be complex and requires further investigation. The manuscript's data does not conclusively explain how RdnI provides a "promiscuous" immunity function, particularly regarding the RdnI mutant/chimera derivatives. The lack of protection observed in these cases might be attributed to other factors, such as a decrease in protein expression levels or misfolding of the proteins. Additionally, the transient nature of the binding interaction could be insufficient to offer effective defenses.

(2) The results from the mixed population competition would benefit from quantitative analysis. The swarm competition assays only yield binary outcomes (Yes or No), limiting the ability to obtain more detailed insights from the data.

(3) The discovery of cross-species protection is solely evident in the heterologous expression-competition model. It remains uncertain whether this is an isolated occurrence or a common characteristic of RdnI immunity proteins across various scenarios. Further investigations are necessary to determine the generality of this behavior.

---

## [Referee Report · Reviewer #4 (Public Review)]

Summary:

Knecht et al. elucidate a Type VI Secretion System (T6SS) effector-immunity pair in Proteus mirabilis. They demonstrate that the effector protein RdnE exhibits DNase activity in vitro and induces toxicity when ectopically expressed in cells, the latter being neutralized by the cognate immunity protein RdnI. The authors identify major regions within RdnI necessary for the interaction and neutralization of RdnE. Notably, they report cross-talk where both cognate and non-cognate RdnI proteins can neutralize RdnE, mitigating its fitness advantage in bacterial co-swarm assays. A comprehensive metagenomic analysis revealed an abundance of rdnI over rdnE genes in most gut samples, suggesting a potential role of rdnI in providing a fitness advantage against bacteria encoding for RdnE effector.

Strengths:

The authors successfully combined biochemical and microbiological experiments with bioinformatics analysis to advance the understanding of the T6SS-mediated population dynamics in bacteria. The co-swarm functional assay is of particular interest as it demonstrates how bacterial strains carrying only rdnI immunity genes could potentially compete in the same niche with other species armed with toxic rdnE effector genes. The manuscript is well-written, and the figures are self-explanatory.

Weaknesses:

(1) How would the authors explain the discrepancy observed in Figure 4 G and Figure 4 S3 B where two RdnI proteins from Prevotella and Pseudomonas genera do not bind to RdnE_Proteus in BACTH, whereas they co-elute with a RdnE_Proteus-FLAG with efficiency comparable to the cross-neutralizing RdnI_Rothia? Similarly, the interaction results obtained in BACTH with RdnI truncate (Figure 4E) or chimeric RdnI (Figure 4I, lane 4) could be a result of an overexpressed T18-fusion variant.

Alternative in vitro protein binding assay would be beneficial.

(2) Based on the bioinformatic analysis the Rothia and Prevotella species harboring rdnE/I genes co-occurred in 5% of metagenomes tested, suggesting that these bacteria could come into contact. The manuscript would benefit greatly if authors demonstrated that RdnI proteins from Rothia or Prevotella could cross-neutralize its own and its 'neighbor' RdnE effectors, for example in an *E. coli* viability assay. The cross-neutralizing co-swarming results (Figure 4F) could also be further validated in viability assay as shown in Figure 2 S1.

(3) Little is known about whether RdnE is delivered via T6SS as a full-length protein or as the shorter C-terminal fragment. There is a possibility that immunity proteins could recognize RdnE regions beyond the C-terminal 138 amino acids that authors used in their in vitro assays.

---

## [Referee Report · Reviewer #5 (Public Review)]

This work investigates a T6SS effector-immunity pair from Proteus mirabilis. The authors make several interesting claims, particularly regarding the mechanism of effector inhibition by the immunity protein. However, it appears that these claims are not fully supported by the evidence provided.

I have read the revised manuscript, the public reviews, and the authors' updated responses to these reviews. In my opinion, the concerns raised by the reviewers remain relevant even after the authors' revisions. Since previous reviews have excellently described the strengths and weaknesses of this work, I will focus on my major concerns:

(1) The authors describe RdnE-RdnI, a T6SS effector-immunity pair from Proteus mirabilis. RdnE is actually the C-terminal domain of IdrD, a 1581-amino-acid protein containing PAAR and RHS domains. This work does not provide evidence for T6SS-dependent secretion of the effector, instead supplying references to previous works.

(2) While the authors claim the function of the RdnE domain is unknown, it was previously shown to be evolutionarily related to PoNe and TseV, both of which are known DNA nucleases. Although the authors cite the relevant references, they do not clearly disclose this information.

(3) The authors claim that RdnE contains two different domains: the first is the PD-(D/E)XK domain, and the second, referred to as "region 2," follows it. Unfortunately, no structural evidence is provided to support this claim, not even a predicted model demonstrating that these are indeed separate domains.

(4) One of the major claims made in this work is that RdnI binding to RdnE is not sufficient for RdnE inhibition, suggesting a more sophisticated mechanism. The authors base this theory on differences between the ability of RdnI to bind RdnE (shown using bacterial two-hybrid assays) and the ability to protect against RdnE toxicity in swarm competition assays. Specifically, they show that the first 85 amino acids of RdnI bind to the short RdnE domain in the bacterial two-hybrid assay but do not protect against the full-length effector in the swarm competition assay. They also demonstrate that performing seven mutations in conserved residues in RdnE or replacing parts of RdnI with parts from other RdnI homologs leads to the same phenomenon.

While these findings are interesting and even intriguing, in my opinion, the current evidence does not support their theory. A simple explanation for the differences between the assays is that while the N-terminal domain of RdnI is sufficient for binding to RdnE, inhibition of the active site of RdnE requires binding of a second domain to RdnE. In that sense, it should be noted that while the authors use co-IP assays to show the interaction between RdnE and full-length RdnI, they do not use it to show the interaction between RdnE and the first 85 amino acids of RdnI.

(5) The authors claim that a "conserved motif" within RdnI plays a role in the inhibition of RdnE. To investigate this, they replace this motif with sequences from several RdnI homologs, demonstrating that in one case, it is possible to exchange these conserved motifs between RdnI homologs that inhibit Proteus RdnE. However, they also show that even if the conserved motif is taken from an RdnI homolog that cannot inhibit Proteus RdnE, the hybrid protein can still protect cells in a swarm competition assay. This result raises concerns regarding the relevance of this conserved motif.

(6) Lastly, regarding the theory that immunity proteins can protect against non-cognate effectors, it appears that the authors based their theory on a single case where RdnI from Rothia protected against RdnE from Proteus. In my opinion, a more thorough investigation, involving testing many homologs, is needed to substantiate this theory.

---

## [Author Response]

The following is the authors’ response to the original reviews.

**eLife assessment**
This work presents valuable information about the specificity and promiscuity of toxic effector and immunity protein pairs. The evidence supporting the claims of the authors is currently incomplete, as there is concern about the methodology used to analyze protein interactions, which did not take potential differences in expression levels, protein folding, and/or transient interaction into account. Other methods to measure the strength of interactions and structural predictions would improve the study. The work will be of interest to microbiologists and biochemists working with toxin-antitoxin and effector-immunity proteins.

We thank the reviewers for considering this manuscript. We agree that this manuscript provides a valuable and cross-discipline introduction to new EI pair protein families where we focus on the EI pair’s flexibility and impacts on community structure. As such, we believe we have provided a solid foundation for future studies to examine non-cognate interactions and their possible effects on microbial communities. This, by definition, leaves some areas “incomplete” and, therefore, open for further investigations. While the methods we show do consider potential differences in binding assays, we have more explicitly addressed how “expression, protein folding, and/or transient binding” may play into this expanded EI pair model. We have also tempered the discussion of the proposed model, while also clearly highlighting other published evidence of non-cognate binding interactions between effector and immunity proteins. We have responded to the reviewers’ public comments (italicized below).

In this revised manuscript, we have updated the main text, particularly the Discussion section, to include more careful language, explain past research better, and add new references to works showing non-cognate immunity proteins protecting against effectors in other systems. We have also updated the supplemental files with more analyses; the relevant procedures are in the Materials and Methods.

**Public Reviews:**

Note: Reviewer 1, who appeared to focus on a subset of the manuscript rather than the whole, based their comments on several inaccuracies, which we discuss below. We found the tone in this reviewer's comments to be, at times, inappropriate, e.g., using "harsh" and "simply too drastic" to imply that common structure-function analyses were outside of the field-standard methods. We also note that the reviewer took a somewhat atypical step in reviewing this manuscript by running and analyzing the potential protein-complex data in AlphaFold2 but did not discuss areas of low confidence within that model that may contradict their conclusions. We are concerned their approach muddled valid scientific criticisms with problematic conclusions.

**Reviewer #1 (Public Review):**
In this manuscript, Knecht, Sirias et al describe toxin-immunity pair from Proteus mirabilis. Their observations suggest that the immunity protein could protect against non-cognate effectors from the same family. They analyze these proteins by dissecting them into domains and constructing chimeras which leads them to the conclusion that the immunity can be promiscuous and that the binding of immunity is insufficient for protective activity.Strengths:The manuscript is well written and the data are very well presented and could be potentially interesting. The phylogenetic analysis is well done, and provides some general insights.Weaknesses:(1) Conclusions are mostly supported by harsh deletions and double hybrid assays. The later assays might show binding, but this method is not resolutive enough to report the binding strength. Proteins could still bind, but the binding might be weaker, transient, and out-competed by the target binding.

The phrasing of structure-function analyses as “harsh” is a bit unusual, as other research groups regularly use deletions and hybrid studies. Given the known caveats to deletion and domain substitutions, we included point-mutation analyses for both the effector and immunity proteins, as found on lines 105 - 113 and 255 - 261 in the current manuscript. These caveats are also why we coupled the in vitro binding analyses with in vivo protection experiments in two distinct experimental systems (*E. coli* and *P. mirabilis*). Based on this manuscript’s introductory analysis (where we define and characterize the genes, proteins, interactions, phylogenetics, and incidences in human microbiomes), the next apparent questions are beyond the scope of this study. Future approaches would include analyzing purified proteins from the effector (E) and immunity (I) protein families using biochemical assays, such as X-ray crystallography, circular dichroism spectroscopy, among others.

Interestingly, most papers in the EI field do not measure EI protein affinity (Jana et al., 2019, Yadav et al., 2021). Notable exceptions are earlier colicin research (Wallis et al., 1995) and a new T6SS EI paper (Bosch et al., 2023) published as we first submitted this manuscript.

(2) While the authors have modeled the structure of toxin and immunity, the toxin-immunity complex model is missing. Such a model allows alternative, more realistic interpretation of the presented data. Firstly, the immunity protein is predicted to bind contributing to the surface all over the sequence, except the last two alpha helices (very high confidence model, iPTM>0.8). The N terminus described by the authors contributes one of the toxin-binding surfaces, but this is not the sole binding site. Most importantly, other parts of the immunity protein are predicted to interact closer to the active site (D-E-K residues). Thus, based on the AlphaFold model, the predicted mechanism of immunization remains physically blocking the active site. However, removing the N terminal part, which contributes large interaction surface will directly impact the binding strength. Hence, the toxin-immunity co-folding model suggests that proper binding of immunity, contributed by different parts of the protein, is required to stabilize the toxin-immunity complex and to achieve complete neutralization. Alternative mechanisms of neutralization might not be necessary in this case and are difficult to imagine for a DNase.

In response to the reviewer’s comment, we again reviewed the RdnE-RdnI AlphaFold2 complex predictions with the most updated version of ColabFold (1.5.2-patch with PDB100 and MMseq2) and have included them at the end of these responses [1].

However, the literature reports that computational predictions of E-I complexes often do not match experimental structural results (Hespanhol et al., 2022, Bosch et al., 2023). As such, we chose not to include the predicted cognate and non-cognate RdnE-I complexes from ColabFold (which uses AlphaFold2) and have not included this data in the revised manuscript. (It is notable that reviewer 1 found the proposed expanded model and research so interesting as to directly input and examine the AI-predicted RdnE-RdnI protein interactions in AlphaFold2.)

Discussion of the prevailing toxin-immunity complex model is in the introduction (lines 45-48) and Figure 5E. Further, there are various known mechanisms for neutralizing nucleases and other T6SS effectors, which we briefly state in the discussion (lines 359 - 361). More in-depth, these molecular mechanisms include active-site blocking (Benz et al., 2012), allosteric-site binding (Kleanthous et al., 1999 and Lu et al., 2014), enzymatic neutralization of the target (Ting et al., 2021), and structural disruption of both the active and binding sites (Bosch et al., 2023). Given this diversity of mechanisms, we did not presume to speculate on the as-of-yet unknown mechanism of RdnI protection. We have expanded discussion of these items in the revised manuscript.

(3) Dissection of a toxin into two domains is also not justified from a structural point of view, it is probably based on initial sequence analyses. The N terminus (actually previously reported as Pone domain in ref 21) is actually not a separate domain, but an integral part of the protein that is encased from both sides by the C terminal part. These parts might indeed evolve faster since they are located further from the active site and the central core of the protein. I am happy to see that the chimeric toxins are active, but regarding the conservation and neutralization, I am not surprised, that the central core of the protein fold is highly conserved. However, "deletion 2" is quite irrelevant - it deletes the central core of the protein, which is simply too drastic to draw any conclusions from such a construct - it will not fold into anything similar to an original protein, if it will fold properly at all.

The reviewer’s comment highlights why we turned to the chimera proteins to dissect the regions of RdnE (formerly IdrD-CT), as the deletions could result in misfolded proteins. (We initially examined RdnE in the years before the launch of AlphaFold2.) However, the reviewer is incorrect regarding the N-terminus of RdnE. The PoNe domain, while also a subfamily of the PD-(D/E)XK superfamily, forms a distinct clade of effectors from the PD-(D/E)XK domain in RdnE (formally IdrD-CT) as seen in Hespanhol et al., 2022; this is true for other DNase effectors as well. Many studies analyzing effectors within the PD-(D/E)XK superfamily only focus on the PD-(D/E)XK domain, removing just this domain from the context of the whole protein (Hespanhol et al., 2022; Jana et al., 2019). Of note, in RdnE, this region alone (containing the DNA-binding domain) is insufficient for DNase activity (unlike in PoNe). We have clarified this distinction in the results section of the current manuscript, visible in figure 2 .

(4) Regarding the "promiscuity" there is always a limit to how similar proteins are, hence when cross-neutralization is claimed authors should always provide sequence similarities. This similarity could also be further compared in terms of the predicted interaction surface between toxin and immunity.

Reviewer 1 points out a fundamental property of protein-protein interactions that has been isolated away from the impacts of such interactions on bacterial community structure. We have provided the whole protein alignments in figure 3 supplemental figure 3, the summary images in Figure 3D, and the protein phylogenetic trees in Figure 3C. We encourage others to consider the protein alignments as percent amino acid sequence similarity is not necessarily a good gauge for protein function and interactions. These data are publicly available on the OSF website associated with this manuscript https://osf.io/scb7z/, and we hope the community explores the data there.

In consideration of the enthusiasm to deeply dive into the primary research data, we have included the pairwise sequence identities across the entire proteins here: Proteus RdnI vs. Rothia RdnI: 23.6%; Proteus RdnI vs. Prevotella RdnI: 16.3%, Proteus RdnI vs. Pseudomonas RdnI: 14.6%; Rothia RdnI vs. Prevotella RdnI: 22.4%, Rothia RdnI vs. Pseudomonas RdnI: 17.6%; Prevotella RdnI vs. Pseudomonas RdnI: 19.5%. (As stated in response to reviewer 1 comment 2, we did not find it appropriate to make inferences based on AlphaFold2-predicted protein complexes.)

Overall, it looks more like a regular toxin-immunity couple, where some cross-reactions with homologues are possible, depending on how far the sequences have deviated. Nevertheless, taking all of the above into account, these results do not challenge toxin-immunity specificity dogma.

In this manuscript, we did not intend to dismiss the E-I specificity model but rather point out its limitations and propose an important expansion of that model that accounts for cross-protection and survival against attacks from other genera. We agree that it is commonly considered that deviations in amino acid sequence over time could result in cross-binding and protection (see lines 364-368). However, the impacts of such cross-binding on community structure, bacterial survival, and strain evolution were rarely addressed in prior literature, with exceptions such as in Zhang et al., 2013 and Bosch et al., 2023 among others. One key insight we propose and show in this manuscript is that cross-binding can be a fitness benefit in mixed communities; therefore, it could be selected for evolutionarily (lines 378-380), even potentially in host microbiomes.

**Reviewer #2 (Public Review):**
Summary:The manuscript by Knecht et al entitled "Non-cognate immunity proteins provide broader defenses against interbacterial effectors in microbial communities" aims at characterizing a new type VI secretion system (T6SS) effector immunity pair using genetic and biochemical studies primarily focused on Proteus mirabilis and metagenomic analysis of human-derived data focused on Rothia and Prevotella sequences. The authors provide evidence that RdnE and RdnI of Proteus constitute an E-I pair and that the effector likely degrades nucleic acids. Further, they provide evidence that expression of non-cognate immunity derived from diverse species can provide protection against RdnE intoxication. Overall, this general line of investigation is underdeveloped in the T6SS field and conceptually appropriate for a broad audience journal. The paper is well-written and, aside from a few cases, well-cited. As detailed below however, there are several aspects of this paper where the evidence provided is somewhat insufficient to support the claims. Further, there are now at least two examples in the literature of non-cognate immunity providing protection against intoxication, one of which is not cited here (Bosch et al PMID 37345922 - the other being Ting et al 2018). In general therefore I think that the motivating concept here in this paper of overturning the predominant model of interbacterial effector-immunity cognate interactions is oversold and should be dialed back.

We agree that analyses focusing on flexible non-cognate interactions and protection are underdeveloped within the T6SS field and are not fully explored within a community structure. These ideas are rapidly growing in the field, as evidenced by the references provided by the reviewer. As stated earlier, we did not intend to overturn the prevailing model but rather have proposed an expanded model that accounts for protection against attacks from foreign genera.

Strengths:One of the major strengths of this paper is the combination of diverse techniques including competition assays, biochemistry, and metagenomics surveys. The metagenomic analysis in particular has great potential for understanding T6SS biology in natural communities. Finally, it is clear that much new biology remains to be discovered in the realm of T6SS effectors and immunity.Weaknesses:The authors have not formally shown that RdnE is delivered by the T6SS. Is it the case that there are not available genetics tools for gene deletion for the BB2000 strain? If there are genetic tools available, standard assays to demonstrate T6SS-dependency would be to interrogate function via inactivation of the T6SS (e.g. by deleting tssC).

Our research group showed that the T6SS secretes RdnE (previously IdrD) in Wenren et al., 2013 (cited in lines 71-73). We later confirmed T6SS-dependent secretion by LC-MS/MS (Saak et al., 2017).

For swarm cross-phyla competition assays (Figure 4), at what level compared to cognate immunity are the non-cognate immunity proteins being expressed? This is unclear from the methods and Figure 4 legend and should be elaborated upon. Presumably these non-cognate immunity proteins are being overexpressed. Expression level and effector-to-immunity protein stoichiometry likely matters for interpretation of function, both in vitro as well as in relevant settings in nature. It is important to assess if native expression levels of non-cognate cross-phyla immunity (e.g. Rothia and Prevotella) protect similarly as the endogenously produced cognate immunity. This experiment could be performed in several ways, for example by deleting the RdnE-I pair and complementing back the Rothia or Prevotella RdnI at the same chromosomal locus, then performing the swarm assay. Alternatively, if there are inducible expression systems available for Proteus, examination of protection under varying levels of immunity induction could be an alternate way to address this question. Western blot analysis comparing cognate to non-cognate immunity protein levels expressed in Proteus could also be important. If the authors were interested in deriving physical binding constants between E and various cognate and non-cognate I (e.g. through isothermal titration calorimetry) that would be a strong set of data to support the claims made. The co-IP data presented in supplemental Figure 6 are nice but are from *E. coli* cells overexpressing each protein and do not fully address the question of in vivo (in Proteus) native expression.

P. mirabilis strain ATCC29906 does not encode the rdnE and rdnI genes on the chromosome (NCBI BioSample: SAMN00001486) (line 151). Production of the RdnI proteins, including the cognate Proteus RdnI, comes from equivalent transgenic expression vectors. Specifically, the rdnI genes were expressed under the flaA promoter in P. mirabilis strain ATCC29906 (Table 1) for the swarm competition assays found in Figure 2C and Figure 4. This promoter results in constitutive expression in swarming cells (Belas et al., 1991; Jansen et al., 2003). In the revised manuscript, figure 4 Supplement Figure 2 shows the relative RdnI protein levels in these strains; we also clarified the expression constructs in the text (see reviewer 3, comment 1).

Lines 321-324, the authors infer differences between E and I in terms of read recruitment (greater abundance of I) to indicate the presence of orphan immunity genes in metagenomic samples (Figure 5A-D). It seems equally or perhaps more likely that there is substantial sequence divergence in E compared to the reference sequence. In fact, metagenomes analyzed were required only to have "half of the bases on reference E-I sequence receiving coverage". Variation in coverage again could reflect divergent sequence dipping below 90% identity cutoff. I recommend performing metagenomic assemblies on these samples to assess and curate the E-I sequences present in each sample and then recalculating coverage based on the exact inferred sequences from each sample.

This comment raises the challenges with metagenomic analyses. It was difficult to balance specificity to a particular species’ DNA sequence with the prevalence of any homologous sequence in the sample. Given the distinction in binding interactions among the examined four species, we opted to prioritize specificity, accepting that we were losing access to some rdnE and rdnI sequences in that decision. We chose a 90% identity cutoff, which, through several in silica controls, ensured that each sequence we identified was the rdnE or rdnI gene from that specific species. For the Version of Record, we have included analysis with a 70% cutoff in the supplemental information to try to account for sequence divergence by lowering the identity cutoffs as suggested. The data from the 70% identity cutoff was consistent with the original data from the 90% identity cutoff.

A description of gene-level read recruitment in the methods section relating to metagenomic analysis is lacking and should be provided.

Noted. We included the raw code and sequences on the OSF website associated with this manuscript https://osf.io/scb7z/.

**Reviewer #3 (Public Review):**
Summary:The authors discovered that the RdnE effector possesses DNase activity, and in competition, P. mirabilis having RdnE outcompetes the null strain. Additionally, they presented evidence that the RdnI immunity protein binds to RdnE, suppressing its toxicity. Interestingly, the authors demonstrated that the RdnI homolog from a different phylum (i.e., Actinomycetota) provides cross-species protection against RdnE injected from P. mirabilis, despite the limited identity between the immunity sequences. Finally, using metagenomic data from human-associated microbiomes, the authors provided bioinformatic evidence that the rdnE/rdnI gene pair is widespread and present in individual microbiomes. Overall, the discovery of broad protection by non-cognate immunity is intriguing, although not necessarily surprising in retrospect, considering the prolonged period during which Earth was a microbial battlefield/paradise.Strengths:The authors presented a strong rationale in the manuscript and characterized the molecular mechanism of the RdnE effector both in vitro and in the heterologous expression model. The utilization of the bacterial two-hybrid system, along with the competition assays, to study the protective action of RdnI immunity is informative. Furthermore, the authors conducted bioinformatic analyses throughout the manuscript, examining the primary sequence, predicted structural, and metagenomic levels, which significantly underscore the significance and importance of the EI pair.Weaknesses:(1) The interaction between RdnI and RdnE appears to be complex and requires further investigation. The manuscript's data does not conclusively explain how RdnI provides a "promiscuous" immunity function, particularly concerning the RdnI mutant/chimera derivatives. The lack of protection observed in these cases might be attributed to other factors, such as a decrease in protein expression levels or misfolding of the proteins. Additionally, the transient nature of the binding interaction could be insufficient to offer effective defenses.

Yes, we agree with the reviewer and hope that grant reviewers’ share this colleague’s enthusiasm for understanding the detailed molecular mechanisms of RdnE-RdnI binding across genera. In the revised manuscript, we have continued to emphasize such caveats as the next frontier is clearly understanding the molecular mechanisms for RdnI cognate or non-cognate protection. In the revised manuscript, figure 4 Supplement Figure 2 shows the RdnI protein levels; we also clarified the expression constructs in the text (see reviewer 2, comment 2).

(2) The results from the mixed population competition lack quantitative analysis. The swarm competition assays only yield binary outcomes (Yes or No), limiting the ability to obtain more detailed insights from the data.

The mixed swam assay is needed when studying T6SS effectors that are primarily secreted during Proteus’ swarming activity (Saak et al. 2017, Zepeda-Rivera et al. 2018). This limitation is one reason we utilize in vitro, in vivo, and bioinformatic analyses. Though the swarm competition assay yields a binary outcome, we are confident that the observed RdnI protection is due to interaction with a trans-cell RdnE via an active T6SS. By contrast, many manuscripts report co-expression of the EI pair (Yadev et al., 2021, Hespanhol et al., 2022) rather than secreted effectors, as we have achieved in this manuscript.

(3) The discovery of cross-species protection is solely evident in the heterologous expression-competition model. It remains uncertain whether this is an isolated occurrence or a common characteristic of RdnI immunity proteins across various scenarios. Further investigations are necessary to determine the generality of this behavior.

We agree, which is why we submitted this paper as a launching point for further investigations into the generality of non-cognate interactions and their potential impact on community structure.

Comments from Reviewing Editor:In addition to the references provided by reviewer#2, the first manuscript to show non-cognate binding of immunity proteins was Russell et al 2012 (PMID: 22607806).IdrD was shown to form a subfamily of effectors in this manuscript by Hespanhol et al 2022 PMID: 36226828 that analyzed several T6SS effectors belonging to PDDExK, and it should be cited.

We appreciate that the reviewer and eLife staff pointed out missed citations. We have incorporated these studies and cited them in the revised manuscript.

[1] The Proteus RdnE in complex with either the Prevotella or Pseudomonas RdnI showed low confidence at the interface (pIDDT ~50-70%); this AI-predicted complex might support the lack of binding seen in the bacterial two-hybrid assay. On the other hand, the Proteus and Rothia RdnI N-terminal regions show higher confidence at the interface with RdnE. Despite this, the C-terminus of the Proteus RdnI shows especially low confidence (pIDDT ~50%) where it might interact near RdnE’s active site (as suggested by reviewer 1). Given this low confidence and the already stated inaccuracies of AI-generated complexes, we would rather wait for crystallization data to inform potential protection mechanisms of RdnI.

**Author response image 1. sa4fig1:**